# ROIDICE: Offline Return on Investment Maximization for Efficient Decision Making

**Woosung Kim**[1*]    **Hayeong Lee**[1*]    **Jongmin Lee**[2†]    **Byung-Jun Lee**[1,3†]

[1]Korea University    [2] UC Berkeley    [3]Gauss Labs Inc.
{wsk208,hayeong_lee,byungjunlee}@korea.ac.kr
jongmin.lee@berkeley.edu
[*]Equal contribution   [†]Corresponding authors

## Abstract

In this paper, we propose a novel policy optimization framework that maximizes Return on Investment (ROI) of a policy using a fixed dataset within a Markov Decision Process (MDP) equipped with a cost function. ROI, defined as the ratio between the return and the accumulated cost of a policy, serves as a measure of the efficiency of the policy. Despite the importance of maximizing ROI in various applications, it remains a challenging problem due to its nature as a ratio of two long-term values: return and accumulated cost. To address this, we formulate the ROI maximizing reinforcement learning problem as linear fractional programming. We then incorporate the stationary distribution correction (DICE) framework to develop a practical offline ROI maximization algorithm. Our proposed algorithm, ROIDICE, yields an efficient policy that offers a superior trade-off between return and accumulated cost compared to policies trained using existing frameworks.

## 1 Introduction

In economics, Return on Investment (ROI) is a financial metric used to evaluate the profitability of an investment relative to its cost. The concept of ROI originates from the work of [5] and is widely regarded as a valuable metric by the majority of marketing managers [4]. ROI is calculated by dividing the profit generated from the investment by the cost of the investment, and it is a key indicator for evaluating the efficiency and effectiveness of various economic decisions and strategies [17]. Maximizing ROI is crucial as a decision with a high ROI yields a higher return at a relatively lower cost.

While ROI is a compelling metric for optimization, ROI maximization for decision making has not been actively explored within the field of machine learning. Previous studies on ROI maximization [3] have focused on multi-armed bandit scenarios, where reward and cost are immediately accessible following a decision. When it comes to sequential decision-making setups like the Markov Decision Process (MDP) that additionally considers a cost function, optimizing a sequence of decisions or a policy to maximize the ratio between return and accumulated cost becomes far more challenging and remains mostly unexplored. This optimization contains two major difficulties: the fractional relationship between the return and accumulated cost of a policy, and the fact that both are long-term quantities collected after a sequence of decisions or actions of the agent. The issues make ROI maximization via existing policy optimization techniques not straightforward, as it includes computing the gradient of the ratio between the two expected values.

For effective optimization of ROI, we refer to the dual formulation of V-LP [16], which represents policy optimization in reinforcement learning (RL) as linear programming in terms of the stationary distribution. The stationary distribution of a policy is the discounted sum of probabilities of the

38th Conference on Neural Information Processing Systems (NeurIPS 2024).

state-action pairs visited by the agent following the policy. Leveraging the property of stationary distribution, which allows for easy estimation of both return and accumulated cost, we derive ROI-LP, linear programming problem for ROI maximization. In this paper, we focus specifically on the offline setting and aim to derive an offline ROI maximization algorithm. The offline setting, which involves optimizing a policy using a fixed dataset of pre-collected experiences without further environmental interactions (e.g., offline RL), has been actively studied recently due to its compelling use cases [2, 7, 9, 10, 11, 20]. Building on the derived ROI-LP, we incorporate a convex regularization to address the distribution shift inherent in offline learning. This leads to our proposed algorithm, ROIDICE, which achieves offline ROI maximization via stationary distribution correction estimation.

Through a series of diverse experiments, we compare ROIDICE to algorithms from other offline policy optimization frameworks, including offline RL and offline constrained RL. We demonstrate that maximizing the ROI of a policy leads to a different, more efficient behavior compared to frameworks that focus solely on maximizing policy return. In summary, our contributions are threefold:

- Formulation of ROI maximization framework in an MDP with a cost function.
- Derivation of ROIDICE, an offline ROI maximization algorithm that optimizes the ROI of a policy using a fixed dataset of pre-collected experiences.
- Comparison with existing offline policy optimization frameworks, showing that ROIDICE optimizes ROI and improves policy efficiency across various domains.

## 2  Backgrounds

**Markov Decision Process with a cost function**  We consider an infinite-horizon discounted Markov Decision Process (MDP) with a cost function, represented as a tuple $\mathcal{M} = \langle S, A, T, r, p_0, \gamma \rangle$. Here, $S$ is the set of states, $A$ is the set of actions, $T(s'|s, a)$ denotes the transition probability from state-action pair $(s, a)$ to the next state $s'$, and $r(s, a)$ is the reward function for state-action pair $(s, a)$. $p_0(s)$ is the initial state distribution, and $\gamma$ is the discount factor. Additionally, we introduce a cost function $c(s, a) > 0$ for all state-action pairs $(s, a)$.

An agent begins at an initial state following $p_0(s)$ and repeats the process of collecting reward $r(s, a)$ and incurring cost $c(s, a)$ by taking action $a$ in state $s$ and transitioning to the next state $s'$. The policy $\pi(a|s)$ represents the distribution of actions to be taken by the agent in state $s$. The performance of the policy $\pi$ can be evaluated with the expected return $R_\pi = \mathbb{E}_\pi \left[ \sum_{t=0}^\infty \gamma^t r(s_t, a_t) \right]$ and the expected accumulated cost $C_\pi = \mathbb{E}_\pi \left[ \sum_{t=0}^\infty \gamma^t c(s_t, a_t) \right]$, which are the expected cumulative sums of discounted rewards and costs over rollouts of the policy $\pi$.

The main goal of reinforcement learning (RL) is to optimize policy $\pi$ to maximize expected return $R_\pi$ collected by the agent. Constrained RL optimizes the policy to maximize its expected return $R_\pi$ while constraining its expected accumulated cost $C_\pi$ to be lower than a given threshold $C_{\text{threshold}}$.

**Stationary Distribution and Linear Programming (LP) formulation of RL**  The stationary distribution of policy $d_\pi$ represents the discounted sum of probabilities of the state-action pair $(s, a)$ visited by an agent following policy $\pi$, i.e., $d_\pi(s, a) := (1 - \gamma) \sum_{t=0}^\infty \gamma^t \Pr(s_t = s, a_t = a)$. Using the stationary distribution, $R_\pi$ and $C_\pi$ can be expressed as linear combinations of the stationary distribution and the reward and the cost respectively: $R_\pi = \frac{1}{1-\gamma} \sum_{s,a} d_\pi(s, a) r(s, a)$ and $C_\pi = \frac{1}{1-\gamma} \sum_{s,a} d_\pi(s, a) c(s, a)$. This characteristic allows the RL problem to be formulated as a linear programming (LP), known as the dual of V-LP in [16].

$$\max_{d \geq 0} \sum_{s,a} d(s, a) r(s, a) \tag{1}$$

$$\text{s.t. } (\mathcal{B}_* d)(s) = (1 - \gamma) p_0(s) + \gamma (\mathcal{T}_* d)(s) \; \forall s \tag{2}$$

where $(\mathcal{B}_* d)(s) = \sum_a d(s, a)$ and $(\mathcal{T}_* d)(s) = \sum_{\bar{s}, \bar{a}} T(s|\bar{s}, \bar{a}) d(\bar{s}, \bar{a})$.

Constraint (2), known as Bellman flow constraint, is a requirement that any stationary distribution must satisfy. The condition can be interpreted as ensuring that the probability of state-action pairs leaving state $s$ matches the total probability of state-action pairs entering $s$, plus the probability of state $s$ initiating a trajectory. Once the optimal stationary distribution $d^*$ is obtained, optimal policy $\pi^*$ can be extracted from it by $\pi^*(a|s) = d^*(s, a) / \sum_{a'} d^*(s, a') \; \forall s, a$.

**Offline RL and DICE-RL Framework** In this paper, we assume an offline policy optimization setting where interaction between the agent and environment is not allowed. Instead, the agent is provided with a fixed dataset of experiences $D = \{(s_i, a_i, r_i, c_i, s_i')\}_{i=1}^N$, consisting of $N$ transition samples, to optimize its policy. A unique challenge in offline RL is balancing between policy optimization and distribution shift, as it is not possible to gather samples from the optimized policy.

In [12], regularized policy optimization is proposed to penalize return maximization (1) with $f$-divergence between stationary distributions of the trained policy $d$ and the behavior policy $d_D$, defined as $D_f(d||d_D) := \sum_{s,a} d_D(s,a)f(d(s,a)/d_D(s,a))$. The convexity of $f$ leads to a convex optimization problem that balances return maximization and distribution shift:

$$\max_{d \geq 0} \sum_{s,a} d(s,a)r(s,a) - \alpha \sum_{s,a} d_D(s,a)f\left(\frac{d(s,a)}{d_D(s,a)}\right)$$

$$\text{s.t. } (\mathcal{B}_* d)(s) = (1-\gamma)p_0(s) + \gamma(\mathcal{T}_* d)(s) \; \forall s$$

where $\alpha$ is a hyper-parameter that controls the trade-off. In the constrained setting studied in [13], a cost constraint $\sum_{s,a} d(s,a)c(s,a) \leq C_{\text{threshold}}$ is added to the problem.

Previous studies have leveraged the characteristics of the above convex optimization problem to develop practical offline RL algorithms. They follow a similar approach by applying a Lagrangian multiplier $\nu(s)$ to the Bellman flow constraint (2), reformulating the Lagrangian dual of the problem into the main loss function of the algorithms. We refer to this family of algorithms as the DICE-RL framework. Detailed explanations of the loss functions and the full derivations of related DICE-RL algorithms (OptiDICE [12] and COptiDICE [13]) are provided in Appendix A.

**Linear-fractional Programming** Linear-fractional programming is defined as maximizing a ratio of two affine functions over a polyhedron. Its standard form is given by:

$$\max_{\mathbf{x}} f(\mathbf{x}) := \frac{\mathbf{c}^T\mathbf{x} + \alpha}{\mathbf{d}^T\mathbf{x} + \beta} \quad \text{s.t. } \mathbf{Gx} \preceq \mathbf{h}, \quad \mathbf{Ax} = \mathbf{b} \tag{3}$$

The domain of $f$ is restricted to ensure the denominator remains positive, i.e., $\{\mathbf{x}|\mathbf{d}^T\mathbf{x} + \beta > 0\}$, as a zero denominator would result in an infeasible solution.

Assuming the region satisfying the constraints is non-empty and bounded, any linear-fractional programming can be transformed into equivalent linear programming using the Charnes-Cooper transformation [1]. The main idea of the transformation is to replace variable $\mathbf{x}$ with two new variables $t \geq 0$ and $\mathbf{y}$, such that $t(\mathbf{d}^T\mathbf{x} + \beta) = 1$ and $\mathbf{y} = t\mathbf{x}$. This change of variables eliminates the denominator of $f(\mathbf{x})$, resulting in the following linear programming problem:

$$\max_{\mathbf{y}, t \geq 0} \mathbf{c}^T\mathbf{y} + \alpha t \quad \text{s.t. } \mathbf{Gy} \preceq t\mathbf{h}, \quad \mathbf{Ay} = t\mathbf{b}, \quad \mathbf{d}^T\mathbf{y} + \beta t = 1.$$

## 3 ROI Maximization in Linear Programming form (ROI-LP)

In this section, we start with a return on investment (ROI) maximization problem by substituting the objective of the dual of V-LP from return $R_\pi$ to ROI. This results in a linear-fractional programming problem, as the objective is the ratio of two linear functions representing return and accumulated cost. We then apply the Charnes-Cooper transformation to derive an equivalent linear programming (LP), ROI-LP.

**ROI Maximization Problem** We begin by defining the ROI of policy $\pi$ in a Markov Decision Process (MDP). The ROI of policy $\pi$ is the ratio between the return $R_\pi$ and its accumulated cost $R_\pi$, which can be expressed in terms of its stationary distribution $d_\pi$: $\text{ROI}(\pi) = \frac{R_\pi}{C_\pi} = \frac{\sum_{s,a} d_\pi(s,a)r(s,a)}{\sum_{s,a} d_\pi(s,a)c(s,a)}$. Substituting the objective of the dual of V-LP (1) with ROI, we get our ROI maximization problem, which adheres to the standard form of linear-fractional programming in (3).

$$\max_{d \geq 0} \frac{\sum_{s,a} d(s,a)r(s,a)}{\sum_{s,a} d(s,a)c(s,a)} \quad \text{s.t. } (\mathcal{B}_* d)(s) = (1-\gamma)p_0(s) + \gamma(\mathcal{T}_* d)(s) \; \forall s. \tag{4}$$

**ROI-LP** We apply the Charnes-Cooper transformation to reformulate the ROI maximization problem (4) into an equivalent linear programming problem, ROI-LP. This involves introducing a non-negative variable $t \geq 0$ and replacing $d(s, a)$ with $d'(s, a)$ and $t$, which are defined as:

$$d'(s, a) = \frac{1}{\sum_{s,a} d(s, a)c(s, a)} d(s, a), \qquad t = \frac{1}{\sum_{s,a} d(s, a)c(s, a)} \tag{5}$$

and thus $d(s, a) = d'(s, a)/t$. Substituting $d$, ROI-LP, the linear programming formulations for maximizing ROI, is given by:

$$\max_{d' \geq 0, t \geq 0} \sum_{s,a} d'(s, a)r(s, a)$$

$$\text{s.t. } (\mathcal{B}_* d')(s) = t(1 - \gamma)p_0(s) + \gamma(\mathcal{T}_* d')(s) \; \forall s$$

$$\sum_{s,a} d'(s, a)c(s, a) = 1 \tag{6}$$

where the equality constraint (6) is newly introduced to ensure that (5) is satisfied. Once the optimization on ROI-LP is complete, optimal stationary distribution $d^*(s, a)$ is obtained by dividing $d'^*(s, a)$ with $t^*$.

*Remark.* When $c(s, a)$ is a fixed constant $C$ for all states and actions, the optimal solution of ROI-LP becomes equivalent to that of the dual of V-LP. In this case, the optimal $t$ becomes $\frac{1}{C}$ as per (5), since any stationary distribution satisfies $\sum_{s,a} d(s, a) = 1$. Consequently, the optimal $d'^*$ becomes $d^*$ from the dual of V-LP scaled by $t$. This special case illustrates that ROI maximization simplifies return maximization when the accumulated cost is constant, regardless of the policy.

## 4 Offline ROI maximization

In this section, we derive an offline ROI maximization algorithm that extends ROI-LP to also include the trade-off for distribution shifts. We formulate a convex optimization problem for offline ROI maximization and use it to derive ROIDICE, the first offline algorithm to optimize policy efficiency in terms of ROI given a fixed dataset.

### 4.1 Regularized ROI Maximization Framework

To formulate an offline ROI maximization problem, we add a convex regularization to ROI-LP representing a distribution shift. The approach is similar to the DICE-RL framework [13], which incorporates an $f$-divergence between the stationary distribution of trained policy $d$ and behavior policy $d_D$ into the dual of V-LP. Our regularized ROI maximization framework is given by:

$$\max_{d' \geq 0, t \geq 0} \sum_{s,a} d'(s, a)r(s, a) - \alpha \sum_{s,a} d_D(s, a)f\left(\frac{d'(s, a)}{d_D(s, a)}, t\right)$$

$$\text{s.t. } (\mathcal{B}_* d')(s) = t(1 - \gamma)p_0(s) + \gamma(\mathcal{T}_* d')(s) \; \forall s \tag{7}$$

$$\sum_{s,a} d'(s, a)c(s, a) = 1 \tag{8}$$

where the design of $f(x, t)$ will be provided later in this subsection. We denote $D_{f,t}(d', d_D) = \sum_{s,a} d_D(s, a)f(\frac{d'(s,a)}{d_D(s,a)}, t)$.

The primary difference between our proposed problem and DICE-RL framework lies in the convex regularization. The $f$-divergence $D_f(d||d_D)$ from DICE-RL framework is not directly applicable to $d'(s, a)$, as $d'(s, a)$ is not a valid probability distribution; it is derived by scaling the stationary distribution $d(s, a)$ by $t$. However, naively incorporating $D_f(\frac{d'}{t}||d_D)$ results in the loss of the convexity, as $f(\frac{x}{t})$ of $f$-divergence is generally not convex with respect to $t$. To this end, we design a new convex regularization that measures the amount of distribution shift while not breaking the convexity of the problem. We begin by outlining three conditions that the regularization should satisfy.

1. $f(x, t)$ should be convex in $x$ given $t > 0$ and convex in $t$ given $x > 0$.

2. $f(x,t)$ should be zero when $x = t$.

3. $f(x,t)$ should be well-defined on $x > 0$ and $t > 0$.

The first condition is that the problem must maintain convexity with respect to both $x$ and $t$. The second condition is that it must effectively regularize the distribution shift. Specifically, $D_{f,t}(d', d_D)$ should be minimized to zero when the stationary distribution $d(s,a) = d'(s,a)/t$ matches $d_D(s,a)$. The last condition is to avoid infeasible optimization scenarios, e.g., negative values inside a $\log$ function. We introduce two convex functions as examples.

$$f_1(x,t) = \frac{1}{2}(x - t)^2, \quad f_2(x,t) = \begin{cases} \frac{1}{2}(x - t)^2 & x \geq t \\ \frac{x}{t}\log(\frac{x}{t}) - \frac{x}{t} + 1 & 0 < x < t \end{cases}$$

From $f(x) = \frac{1}{2}(x - 1)^2$ of $\chi^2$-divergence, we can obtain the first function $f_1(x,t)$ by shifting it by $t - 1$. We introduce a softened variant of $f_1$, denoted as $f_2$, where a scaled version of the function $f(x) = x\log x - x + 1$ (arising from the KL divergence) is seamlessly combined with the left segment of $f_1$. Although $f_2(x,t)$ is not entirely convex with respect to $t$, it serves as a valid regularization term, as $f_2(x,t) = 0$ holds exclusively when $x = t$.

*Remark.* A unique characteristic of the proposed regularization $D_{f,t}(d', d_D)$ is that its magnitude is affected not only by the degree of distribution shift but also by the trained policy's accumulated cost. We demonstrate this by comparing the magnitude of $D_{f,t}(d', d_D)$ and $f$-divergence $D_f(\frac{d'}{t} \| d_D)$ that corresponds to typical regularizer of DICE-RL framework. In the case of $\chi^2$-divergence, $D_{f,t}(d', d_D) = t^2 D_f(\frac{d'}{t} \| d_D)$ as $D_f(\frac{d'}{t} \| d_D) = \sum_{s,a} d_D(s,a)\frac{1}{2t^2}(\frac{d'(s,a)}{d_D(s,a)} - t)^2$. Since $t$ represents the inverse of the expected accumulated cost (5), the strength of the regularization with $D_{f,t}(d', d_D)$ becomes stronger than the actual $f$-divergence when the expected accumulated cost of the trained policy is less than 1.

## 4.2 ROIDICE

We are now prepared to derive ROIDICE, an offline algorithm for ROI maximization based on the Regularized ROI maximization framework. To enforce the constraints (7) and (8), we introduce Lagrangian multiplier $\nu \in \mathbb{R}^{|S|}$ and $\mu \in \mathbb{R}$:

$$\max_{w \geq 0, t \geq 0} \min_{\nu, \mu} L(\nu, \mu, w, t) := \sum_{s,a} w(s,a)d_D(s,a)r(s,a) - \alpha \sum_{s,a} d_D(s,a)f(w(s,a), t) \tag{9}$$

$$+ \sum_s \nu(s)\left(t(1 - \gamma)p_0(s) + \gamma(\mathcal{T}_* d')(s) - (\mathcal{B}_* d')(s)\right) + \mu\left(1 - \sum_{s,a} w(s,a)d_D(s,a)c(s,a)\right),$$

where $w(s,a) = d'(s,a)/d_D(s,a)$. We follow a similar derivation process as outlined by [12] and obtain a closed-form solution of $w^*_{\nu,\mu,t}(s,a)$:

$$w^*_{\nu,\mu,t}(s,a) = \max\left(0, (f')^{-1}\left(\frac{e_\nu(s,a) - \mu c(s,a)}{\alpha}, t\right)\right) \quad \forall s, a. \tag{10}$$

where $e_\nu(s,a) = r(s,a) + \gamma \sum_{s'} T(s'|s,a)\nu(s') - \nu(s)$ and $(f')^{-1}(y,t)$ is the inverse function of $\frac{\partial}{\partial x}f(x,t)$ with respect to $x$. Loss functions of $\nu, \mu$ and $t$ are obtained by substituting $w^*_{\nu,\mu,t}(s,a)$ into $L(\nu, \mu, w, t)$:

$$\min_\nu \mathbb{E}_{s \sim p_0}[t(1 - \gamma)\nu(s)] + \mathbb{E}_{(s,a) \sim d_D}\left[w^*_{\nu,\mu,t}(s,a)(e_\nu(s,a) - \mu c(s,a)) - \alpha f(w^*_{\nu,\mu,t}(s,a), t)\right],$$

$$\min_\mu \mathbb{E}_{(s,a) \sim d_D}\left[w^*_{\nu,\mu,t}(s,a)(e_\nu(s,a) - \mu c(s,a)) - \alpha f(w^*_{\nu,\mu,t}(s,a), t)\right] + \mu,$$

$$\max_{t \geq 0} \mathbb{E}_{s \sim p_0}[t(1 - \gamma)\nu(s)] - \mathbb{E}_{(s,a) \sim d_D}\left[\alpha f(w^*_{\nu,\mu,t}(s,a), t)\right].$$

After the minimization on $\nu, \mu$ and $t$ is complete, we apply policy extraction techniques from [12] to obtain the policy that generates optimal stationary distribution $d^*(s,a) = \frac{1}{t}w^*_{\nu,\mu,t}(s,a)d_D(s,a)$. In this paper, we adopt weighted behavior cloning method, whose loss function is given as,

$$\max_\theta \mathbb{E}_{(s,a) \sim d_D}\left[\frac{1}{t}w^*_{\nu,\mu,t}(s,a)\log \pi_\theta(a|s)\right] \tag{11}$$

The more detailed derivation and algorithm of ROIDICE can be found in Appendix B.

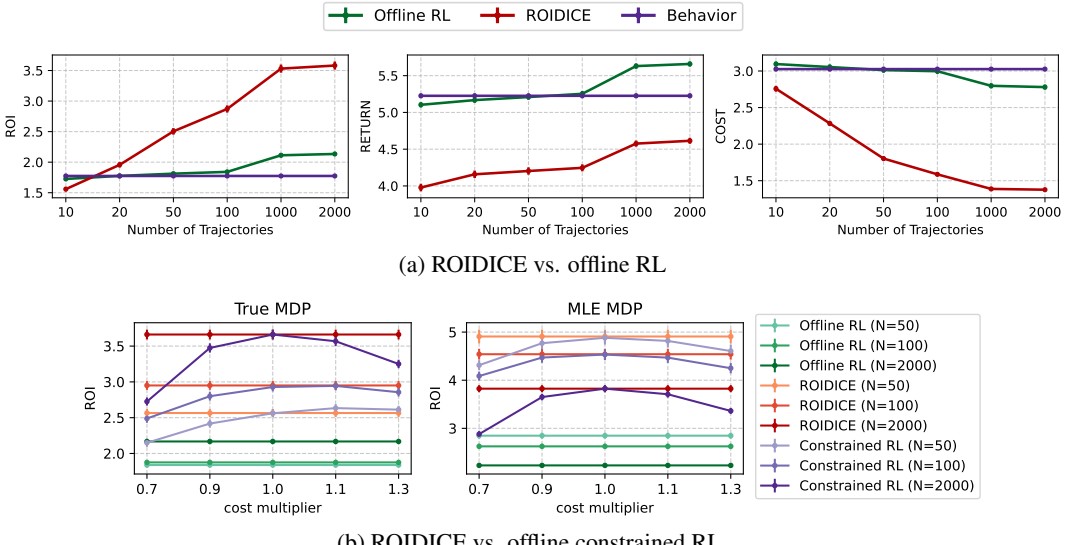

(a) ROIDICE vs. offline RL

(b) ROIDICE vs. offline constrained RL.

Figure 1: Comparison of ROIDICE with other offline algorithms. We average the scores and obtain $\pm 2\times$ standard error using 1000 seeds. $N$ denotes the number of trajectories within the dataset.

## 5 Experiments

In this section, we demonstrate that the offline policy obtained from ROIDICE achieves a superior trade-off between return and accumulated cost compared to other RL approaches such as offline RL and constrained offline RL, in both finite and continuous domains.

### 5.1 Random Finite MDP Experiments

We test ROIDICE in a Random MDP setting similar to [12]. A tabular MDP with $|S| = 50$ and $|A| = 4$ is randomly generated and a behavior policy that achieves $90\%$ of the optimal return is used to gather fixed datasets with $\{10, 20, 50, 100, 1000, 2000\}$ trajectories. We estimate the empirical transition probability $\widehat{T}(s'|s, a)$ with the given samples and use it to run a tabular version of ROIDICE. Details of the experiments are provided in the Appendix C. We compare ROIDICE with DICE-RL on other offline policy optimization schemes, OptiDICE [12] for offline RL and COptiDICE [13] for offline constrained RL.

**vs. Offline RL** In Figure 1a, we compare the performance of ROIDICE with OptiDICE. While OptiDICE achieves the highest return, it incurs a higher accumulated cost than ROIDICE. This is expected, as RL does not account for the accumulated cost it incurs. In contrast, ROIDICE optimizes the policy's ROI, resulting in a highly efficient policy that delivers a substantial return at a relatively lower accumulated cost.

**vs. Offline Constrained RL** In Figure 1b, we compare the performance of ROIDICE with COptiDICE. Offline constrained RL aims to maximize return while keeping the accumulated cost below a specified threshold $C_\pi \leq C_{\text{threshold}}$. As demonstrated in Appendix A, COptiDICE optimizes its policy using a penalized reward function $r(s, a) - \lambda_c c(s, a)$, where $\lambda_c$ is the Lagrange multiplier for the cost constraint. To ensure satisfaction of the cost constraint, $\lambda_c$ is updated to increase when the constraint is violated. To set the cost thresholds for our experiment, we obtain the accumulated cost of the ROIDICE policy on the Maximum Likelihood Estimate (MLE) MDP and multiply it with $[0.7, 0.9, 1.0, 1.1, 1.3]$. The left side of the figure shows the true ROI performance of the agents, while the right side shows the ROI performance on the MLE MDP, representing the performance agents expect. The cost thresholds from the MLE MDP are based on the offline assumption which excludes the interaction with the true MDP.

With a sufficient number of trajectories in the offline dataset, COptiDICE with a cost multiplier of 1.0, which corresponds to the same cost budget as the optimized ROIDICE policy, achieves the best ROI among the tested multipliers. Both increasing or decreasing the cost threshold result in reduced

Table 1: ROI of ROIDICE compared with offline RL and offline constrained RL algorithms. We average each score and get $\pm 2\times$ standard error with 5 seeds across 10 episodes. The task name is succinctly stated: Hopper (H), Walker2D (W), Halfcheetah (HC), and Finance (F).

| Task | ROIDICE | OptiDICE | COptiDICE | | CDT | |
|---|---|---|---|---|---|---|
| | | | 50th | 80th | 50th | 80th |
| H-m | **9.21±0.49** | 3.58±0.03 | 7.37±0.19 | 7.21±0.15 | 3.04±0.2 | 3.0±0.16 |
| H-m-e | **8.29±0.18** | 4.93±0.36 | 7.6±0.44 | 7.88±0.21 | 3.71±0.15 | 3.73±0.16 |
| H-e | **8.51±0.25** | 5.23±0.04 | 7.9±0.14 | 7.99±0.18 | -0.1±0.17 | -0.17±0.23 |
| W-m | **3.06±0.14** | 2.34±0.15 | 2.66±0.63 | 2.92±0.48 | 0.6±0.4 | 0.82±0.31 |
| W-m-e | **4.21±0.78** | 4.03±0.06 | 3.45±0.57 | 3.3±0.49 | 0.91±0.49 | 0.77±0.35 |
| W-e | **5.01±0.42** | 4.59±0.02 | 4.25±0.12 | 4.2±0.12 | -0.01±0.2 | -0.01±0.2 |
| HC-m | **8.48±0.19** | 6.18±0.05 | 8.04±0.4 | 7.98±0.32 | -0.03±0.01 | -0.01±0.01 |
| HC-m-e | **10.17±0.4** | 9.04±1.03 | 8.31±0.4 | 8.06±0.47 | 6.12±0.78 | 6.02±0.08 |
| HC-e | 12.69±0.09 | **12.74±0.13** | 7.42±0.86 | 7.91±0.39 | 12.67±0.07 | 12.66±0.06 |
| F-M | **43.2±13.13** | 40.3±10.29 | 24.4±8.5 | 16.3±5.84 | 20.5±12.63 | 20.5±12.63 |
| F-H | **47.2±8.67** | 41.8±7.23 | 27.3±6.41 | 26.4±4.31 | 46.5±8.97 | 46.5±8.98 |

policy efficiency, as constrained RL agents typically use the entire available budget to maximize return. When the number of trajectories is insufficient, ROIDICE is rarely outrun by COptiDICE as the large discrepancy between True MDP and MLE MDP causes ROIDICE to be optimized on the erroneous MLE MDPs.

While it is theoretically possible to identify a policy with optimal ROI using a constrained RL algorithm, doing so requires multiple runs with different cost thresholds to determine the best one. This process can become computationally intractable and less accurate as the complexity of the problem increases. In contrast, a single run of ROIDICE is sufficient to find a policy with high ROI without the need to explicitly search for the right cost threshold.

## 5.2 ROI Maximization in Continuous Domains

We evaluate our algorithm's performance across diverse domains using datasets from D4RL [6] (CC BY 4.0) and NeoRL [18] (CC BY 4.0). From the D4RL benchmark, we select three locomotion tasks (Hopper, Halfcheetah, Walker2D), and from NeoRL, we include a stock trading task (FinRL). To assess ROIDICE's capability to maximize Return on Investment (ROI), we conduct comparative analyses with various unconstrained or constrained offline RL algorithms. We utilize offline datasets from D4RL and NeoRL, which are collected using data collection policies that do not consider cost. Similar to the random MDP experiments, we include OptiDICE and COptiDICE for comparison. Additionally, we incorporate the Constrained Decision Transformer (CDT, [15]) to represent a recent approach that leverages the power of generative models on RL.

### 5.2.1 Environment and Offline Dataset

In the three locomotion environments, observations include the positional values and velocities of various body parts, with actions corresponding to the torques exerted between links. The common reward function used in recent benchmarks consists of three components: incentive for forward running speed, incentive for maintaining a healthy state, and penalty for applying torques on joints. In this experiment, we use the control penalty as the cost function, aiming to train the agent to strive for efficient energy consumption from an ROI perspective. We used only forward running speed as our reward function.

FinRL, the time-driven stock trading simulator, enables us to replicate live stock markets using real market data. This allows us to assess our algorithm's proficiency in both resource management and decision-making amidst uncertainty. Observations include balances, daily stock amounts, closing

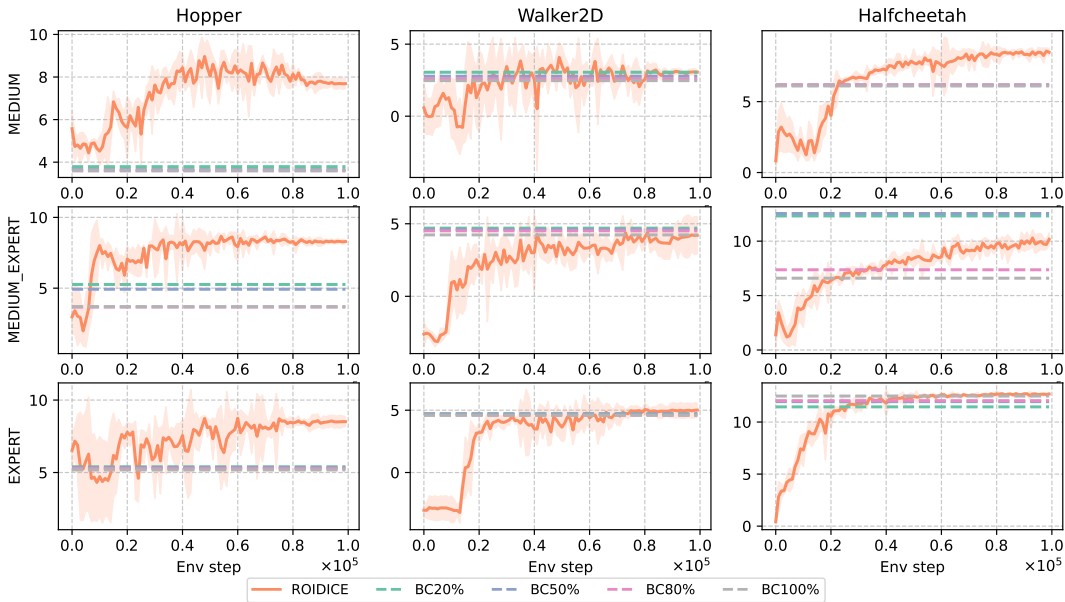

Figure 2: ROI Comparison of ROIDICE and Dataset with varying dataset qualities. We average the each scores and get $\pm 2\times$ standard error with 5 seeds across 10 episodes. BC$n$% refers to behavior cloning utilizing the top $n$% of the offline dataset, ranked by ROI.

prices, and various technical indicators for each of the 30 stocks in the trading pool. Agents use their balances to trade these stocks. We define a cost function based on trading volume and stock price, imposing a fee on each transaction. This requires the agent to strategically optimize its transactions, considering both stock price fluctuations and associated fees. For more details on each environment and experimental setting, see Appendix E.

### 5.2.2 Results

Table 1 presents our results for all four tasks. Each algorithm is trained for 100k steps and evaluated for a maximum of 1000 steps with 5 seeds across 10 episodes. The data quality levels are denoted as follows: for locomotion tasks, medium (m), medium-expert (m-e), and expert (e); for the financial task, medium (M), and high (H). Complete experimental results are provided in Appendix H.

ROIDICE exhibits superior ROI performance compared to other algorithms across most tasks, particularly excelling in the Hopper environment where there is significant potential for ROI improvement. OptiDICE also demonstrates high ROI in environments other than Hopper due to the high ROI of the offline datasets themselves for Walker2D and Halfcheetah (see Figure 2 for dataset ROIs). However, compared to OptiDICE, ROIDICE shows a greater return with lower accumulated cost in a significant number of tasks (see Appendix H).

The constrained offline RL algorithms are evaluated under two cost thresholds, specifically using the 80th and 50th percentiles of the accumulated costs from the offline dataset to set the cost conditions for COptiDICE (discounted with $\gamma = 0.99$) and CDT (undiscounted). This constraint setting ensures that the constrained offline RL algorithms make in-distribution decisions regarding cost constraints. In some domains, particularly where costs can be lowered with a slight loss in return, COptiDICE demonstrates a higher ROI compared to OptiDICE, but without fine-grained tuning of cost constraints, it does not reach the performance of ROIDICE. Conversely, CDT tends to overfit under the current experiment settings when the dataset lacks diversity. In most cases, CDT either fails to optimize a policy (resulting in negative ROI), or optimizes a policy with a very high return but also a very high accumulated cost (resulting in a small ROI).

Figure 2 illustrates how ROIDICE effectively learns from diverse behaviors in the dataset. We compare the ROI of ROIDICE with that of Behavior Cloning (BC) that clones behaviors only from high ROI trajectories. We compute the ROI of trajectories in the dataset and select the top 20%, 50%, and 80% ROI trajectories to train BC$n$% agents. We can observe that in a significant number of experiment settings, ROIDICE outperforms BC$n$% agents. This demonstrates ROIDICE's capability

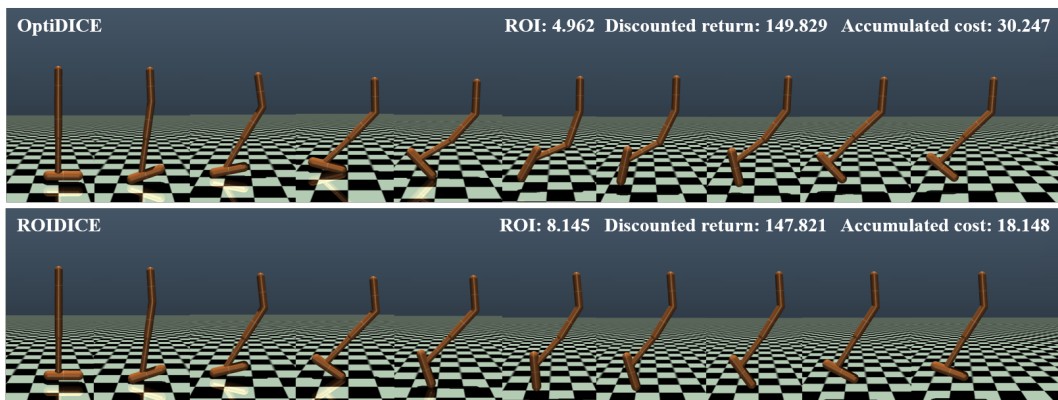

Figure 3: Visualization of ROIDICE and OptiDICE in Hopper environment.

to stitch together various segments from different trajectories to optimize a policy that behaves more efficiently than any other trajectory in the dataset.

Compared to locomotion tasks, the financial task adds another challenge of high stochasticity. Despite this, ROIDICE demonstrates strong performance on the financial task with both medium and high-quality datasets, as shown in Table 1. OptiDICE, although achieving high returns, shows relatively low ROI performance due to high transaction fees (cost). The high ROI observed from CDT is actually coincidental, as CDT fails to meet the given cost constraint due to overfitting; its return maximization policy accidentally results in an efficient policy with high ROI.

**Qualitative Behavior Examples** Figure 3 illustrates the different behaviors between ROIDICE and OptiDICE in the Hopper task at the same time steps, selected to clearly represent these differences. Recall that in the locomotion task, the cost corresponds to the amount of torque applied, i.e., the energy consumed forcing it to move forward. OptiDICE focuses solely on maximizing return, resulting in the agent making large, far-jumping movements, even applying torque to the joint after the jump, which is inefficient for obtaining additional rewards. In contrast, ROIDICE finds the optimal behavior to increase return while reducing energy consumption. Consequently, the agent attempts to hop far but avoids inefficient actions that decrease ROI.

## 6 Conclusion

In this paper, we propose a novel policy optimization framework designed to maximize the Return on Investment (ROI) of the policy, the ratio between the return and the accumulated cost. Maximizing ROI is particularly relevant in practical scenarios: the fuel efficiency (km/L) of autonomous vehicles, managing the trade-off between decision quality and planning time in meta-controllers for real-time decision-making, or balancing generation quality and inference time in large language models (LLMs). The ROI maximization is approached through linear-fractional programming that incorporates the stationary distribution of the policy. We apply the Charnes-Cooper transformation to convert the problem into equivalent linear programming, ROI-LP. Motivated by the DICE RL frameworks, we extend ROI-LP to develop our offline policy optimization algorithm, ROIDICE, by incorporating convex regularization designed to properly regularize the distribution shift. We show that ROIDICE yields a policy with better efficiency than policies from the existing RL-based optimization methods by considering the trade-off between return and accumulated cost.

## 7 Limitations

While our work is innovative in optimizing the ROI of a policy, it functions within the constraints of an offline setting where direct interaction with the environment is not feasible. Consequently, the ultimate performance of the policy optimized with ROIDICE is contingent upon the quality of the provided dataset. Since ROIDICE takes into account both returns and accumulated costs, it is sensitive to the design of the reward and cost functions. For instance, the agent may behave similarly

to one trained by an unconstrained RL when the cost function approximates a constant. Conversely, the agent may exhibit behaviors aimed at minimizing costs excessively, akin to laziness, when the cost function varies significantly.

## Acknowledgements

This work was partly supported by Institute of Information & communications Technology Planning & Evaluation (IITP) grant funded by the Korea government (MSIT) (No. RS-2022-II220311, Development of Goal-Oriented Reinforcement Learning Techniques for Contact-Rich Robotic Manipulation of Everyday Objects, No. RS-2024-00457882, AI Research Hub Project, and No. RS-2019-II190079, Artificial Intelligence Graduate School Program(Korea University)), the IITP(Institute of Information & Coummunications Technology Planning & Evaluation)-ITRC(Information Technology Research Center)(IITP-2024-RS-2024-00436857) grant funded by the Korea government(Ministry of Science and ICT), the NRF (RS-2024-00451162) funded by the Ministry of Science and ICT, Korea, BK21 Four project of the National Research Foundation of Korea, and NSF AI4OPT AI Centre.

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

# A  Full derivation of DICE RL algorithms

In this section, we introduce how DICE RL algorithms are derived from the regularized policy optimization framework. We derive DICE RL algorithms for both unconstrained and constrained scenario: OptiDICE [12] and COptiDICE [13].

## A.1  Derivation of OptiDICE

We show how OptiDICE is derived from the regularized policy optimization framework in [12]. The convex optimization of the framework is given as,

$$\max_{d \geq 0} \sum_{s,a} d(s,a)r(s,a) - \alpha \sum_{s,a} d_D(s,a)f\left(\frac{d(s,a)}{d_D(s,a)}\right)$$

$$\text{s.t.} \sum_a d(s,a) = (1-\gamma)p_0(s) + \gamma \sum_{\bar{s},\bar{a}} T(s|\bar{s},\bar{a})d(\bar{s},\bar{a}) \ \forall s$$

where convex function $f(x)$ is a convex function generally used in $f$-divergence: $f(x) = \frac{1}{2}(x-1)^2$ of $\chi^2$-divergence or $f(x) = x\log x - x + 1$ of KL divergence.

Assuming $d(s,a) = w(s,a)d_D(s,a)$ is the stationary distribution of the dataset $d_D(s,a)$ weighted by $w$, the Lagrangian dual of the convex optimization problem is given as,

$$\max_{w \geq 0} \min_{\nu} \ L(w,\nu) := \sum_{s,a} w(s,a)d_D(s,a)r(s,a) - \alpha \sum_{s,a} d_D(s,a)f(w(s,a)) \tag{12}$$

$$+ \sum_s \nu(s)\left[(1-\gamma)p_0(s) + \gamma \sum_{\bar{s},\bar{a}} T(s|\bar{s},\bar{a})w(\bar{s},\bar{a})d_D(\bar{s},\bar{a}) - \sum_a w(s,a)d_D(s,a)\right]$$

where $\nu$ is the Lagrangian multiplier for the Bellman flow constraint.

We use $\sum_s \nu(s)\sum_{\bar{s},\bar{a}} T(s|\bar{s},\bar{a})w(\bar{s},\bar{a})d_D(\bar{s},\bar{a}) = \sum_{s,a} w(s,a)d_D(s,a)\sum_{s'} T(s'|s,a)\nu(s')$ to switch the order of summation and transform the Lagrangian.

$$\max_{w \geq 0} \min_{\nu} \ L(w,\nu) := (1-\gamma)\mathbb{E}_{s \sim p_0}[\nu(s)] + \mathbb{E}_{(s,a) \sim d_D}[w(s,a)e_\nu(s,a) - \alpha f(w(s,a))]$$

where $e_\nu(s,a) = r(s,a) + \gamma \sum_{s'} T(s'|s,a)\nu(s') - \nu(s)$.

OptiDICE [12] simplifies the max-min problem by switching the order of optimization from $\max_{w \geq 0} \min_\nu L(w,\nu)$ to $\min_\nu \max_{w \geq 0} L(w,\nu)$. The optimal solution is not affected due to the strong duality of the problem. The switch enables the closed-form solution of stationary distribution ratio $w$ of the inner maximization problem that satisfies $\partial L(w,\nu)/\partial w(s,a) = 0 \ \forall s,a$.

$$w_\nu^*(s,a) = \max\left(0, (f')^{-1}\left(\frac{e_\nu(s,a)}{\alpha}\right)\right) \tag{13}$$

The closed-form solution $w_\nu^*(s,a)$ can be plugged in $L(w,\nu)$ to derive $\nu$ loss of OptiDICE given as,

$$\min_\nu \ L(w_\nu^*,\nu) = (1-\gamma)\mathbb{E}_{s \sim p_0}[\nu(s)] + \mathbb{E}_{(s,a) \sim d_D}[w_\nu^*(s,a)e_\nu(s,a) - \alpha f(w_\nu^*(s,a))] \tag{14}$$

After minimization on $\nu$ is complete, optimal $\nu^*$ is plugged in $w_\nu^*(s,a)$ to obtain the optimal stationary distribution $d^*(s,a)$. Policy $\pi$ that generates $d^*(s,a)$ is trained by using policy extraction techniques from [12]. In this paper, we adopt weighted behavior cloning method, whose loss function is given as,

$$\max_\theta \mathbb{E}_{(s,a) \sim d_D}[w_{\nu^*}^*(s,a) \log \pi_\theta(a|s)] \tag{15}$$

## A.2  Derivation of COptiDICE

We show how COptiDICE, the extension of OptiDICE to offline constrained RL, is derived. Constrained RL assumes an additional cost function $c(s,a)$ and limits the discounted sum of cost

$\sum d(s,a)c(s,a) \leq C_{\text{threshold}}$. The constraint is added to the regularized policy optimization framework in [13].

$$\max_{d \geq 0} \sum_{s,a} d(s,a)r(s,a) - \alpha \sum_{s,a} d_D(s,a)f\left(\frac{d(s,a)}{d_D(s,a)}\right)$$

$$\text{s.t.} \sum_{a} d(s,a) = (1-\gamma)p_0(s) + \gamma \sum_{\bar{s},\bar{a}} T(s|\bar{s},\bar{a})d(\bar{s},\bar{a}) \; \forall s$$

$$\sum_{s,a} d(s,a)c(s,a) \leq C_{\text{threshold}}$$

Lagrangian multiplier $\lambda$ is applied to the cost constraint and the Lagrangian of the constrained problem is given as,

$$\max_{w \geq 0} \min_{\nu,\lambda} L_c(w,\nu) := L(w,\nu) + \lambda(C_{\text{threshold}} - \sum_{s,a} w(s,a)d_D(s,a)c(s,a)) \tag{16}$$

After the similar derivations from OptiDICE, the closed-form solution of stationary distribution ratio $w$ and loss functions for $\nu$ are given as,

$$w_{\nu,\lambda}^*(s,a) = \max\left(0, (f')^{-1}\left(\frac{e_{\nu,\lambda}(s,a)}{\alpha}\right)\right) \tag{17}$$

$$\min_{\nu} (1-\gamma)\mathbb{E}_{s \sim p_0}[\nu(s)] + \mathbb{E}_{(s,a) \sim d_D}[w_{\nu,\lambda}^*(s,a)e_{\nu,\lambda}(s,a) - \alpha f(w_{\nu,\lambda}^*(s,a))] \tag{18}$$

where $e_{\nu,\lambda}(s,a) := r(s,a) - \lambda c(s,a) + \sum_{s'} T(s'|s,a)\nu(s') - \nu(s)$. Lagrangian multiplier $\lambda$ determines how much reward function $r(s,a)$ is penalized by cost function $c(s,a)$ and its loss function is given as,

$$\max_{\lambda} \lambda\left(\sum_{s,a} w_{\nu,\lambda}^*(s,a)d_D(s,a)c(s,a) - C_{\text{threshold}}\right) \tag{19}$$

where $\lambda$ is updated to satisfy the constraint by increasing when discounted sum of cost $\sum_{s,a} d(s,a)c(s,a)$ is larger than the cost upper bound $C_{\text{threshold}}$.

## B  Derivation of ROIDICE

We follow similar approaches from Appendix A to derive ROIDICE from the Regularized ROI maximization framework. The main goal of the framework is to obtain the optimal stationary distribution $d^*(s,a) = d'^*(s,a)/t^*$ that maximizes ROI of the policy from the optimal solutions of the convex optimization optimization given as,

$$\max_{d' \geq 0, t \geq 0} \sum_{s,a} d'(s,a)r(s,a) - \alpha \sum_{s,a} d_D(s,a)f\left(\frac{d'(s,a)}{d_D(s,a)}, t\right)$$

$$\text{s.t.} (\mathcal{B}_* d')(s) = t(1-\gamma)p_0(s) + \gamma(\mathcal{T}_* d')(s) \; \forall s$$

$$\sum_{s,a} d'(s,a)c(s,a) = 1$$

We derive the Lagrangian of the convex problem by introducing Lagrangian multipliers $\nu$ and $\mu$ for the equality constraints. We change the optimization variable $d'(s,a)$ to $w(s,a) = d'(s,a)/d_D(s,a)$, but the ratio is not a valid stationary distribution ratio in contrast to the previous DICE RL algorithms.

$$\max_{w \geq 0, t \geq 0} \min_{\nu,\mu} L(\nu,\mu,w,t) := \sum_{s,a} w(s,a)d_D(s,a)r(s,a) - \alpha \sum_{s,a} d_D(s,a)f(w(s,a),t)$$

$$+ \sum_{s} \nu(s)\left(t(1-\gamma)p_0(s) + \gamma(\mathcal{T}_* d')(s) - (\mathcal{B}_* d')(s)\right) + \mu\left(1 - \sum_{s,a} w(s,a)d_D(s,a)c(s,a)\right)$$

We use $\sum_s \nu(s) \left(\gamma(\mathcal{T}_* d')(s) - (\mathcal{B}_* d')(s)\right) = \sum_{s,a} d'(s,a) \left(\gamma \sum_{s'} T(s'|s,a)\nu(s') - \nu(s)\right)$ to obtain an another expression of $L(\nu, \mu, w, t)$.

$$\max_{w\geq 0, t\geq 0} \min_{\nu,\mu} L(\nu, \mu, w, t) = \mathbb{E}_{s_0 \sim p_0} \left[t(1-\gamma)\nu(s_0)\right] + \mathbb{E}_{(s,a)\sim d_D}[w(s,a)(e_\nu(s,a) - \mu c(s,a))$$
$$- \alpha f(w(s,a), t)] + \mu$$

where $e_\nu(s,a) = r(s,a) + \gamma \sum_{s'} T(s'|s,a)\nu(s') - \nu(s)$.

We follow the approaches from OptiDICE [12] and switch the order of the optimization from $\max_{w\geq 0, t\geq 0} \min_{\nu,\mu} L(\nu, \mu, w, t)$ to $\max_{t\geq 0} \min_{\nu,\mu} \max_{w\geq 0} L(\nu, \mu, w, t)$ to obtain the closed-form solution of $w$. Optimal $w$ that satisfies $\frac{\partial L(\nu,\mu,w,t)}{\partial w(s,a)} = 0$ in terms of $\nu, \mu$ and $t$ is given as,

$$w^*_{\nu,\mu,t}(s,a) = \max\left(0, (f')^{-1}\left(\frac{e_\nu(s,a) - \mu c(s,a)}{\alpha}, t\right)\right) \tag{20}$$

where $(f')^{-1}(y,t)$ is an inverse function of $\frac{\partial}{\partial x} f(x,t)$ with respect to $x$. Loss functions of $\nu, \mu$ and $t$ are obtained by substituting $w^*_{\nu,\mu,t}(s,a)$ into $L(\nu, \mu, w, t)$:

$$\min_\nu \mathcal{L}_{\nu_\phi} := \mathbb{E}_{s\sim p_0}\left[t(1-\gamma)\nu(s)\right] \tag{21}$$
$$+ \mathbb{E}_{(s,a)\sim d_D}\left[w^*_{\nu,\mu,t}(s,a)(e_\nu(s,a) - \mu c(s,a)) - \alpha f(w^*_{\nu,\mu,t}(s,a), t)\right],$$
$$\min_\mu \mathcal{L}_\mu := \mathbb{E}_{(s,a)\sim d_D}\left[w^*_{\nu,\mu,t}(s,a)(e_\nu(s,a) - \mu c(s,a)) - \alpha f(w^*_{\nu,\mu,t}(s,a), t)\right] + \mu,$$
$$\max_{t\geq 0} \mathcal{L}_t := \mathbb{E}_{s\sim p_0}\left[t(1-\gamma)\nu(s)\right] - \mathbb{E}_{(s,a)\sim d_D}\left[\alpha f(w^*_{\nu,\mu,t}(s,a), t)\right].$$

---

**Algorithm 1** ROIDICE

---

**Input**: The offline dataset $D$, the initial state offline dataset $p_0$, parameterized Lagrangian multipliers $\nu_\phi, \mu, t$, and policy $\pi_\theta$.
**Output**: Optimal policy $\pi_\theta^*$.
Initialize all parameters.
**while** convergence **do**
    Update $\nu_\phi, \mu, t$ with $\mathcal{L}_{\nu_\phi}, \mathcal{L}_\mu, -\mathcal{L}_t$
    Update $\theta$ with $\mathcal{L}_\theta = -\mathbb{E}_{(s,a)\sim D}\left[\frac{1}{t} \cdot w^*_{\nu,\mu,t}(s,a) \cdot \log \pi_\theta(a|s)\right]$
**end while**

---

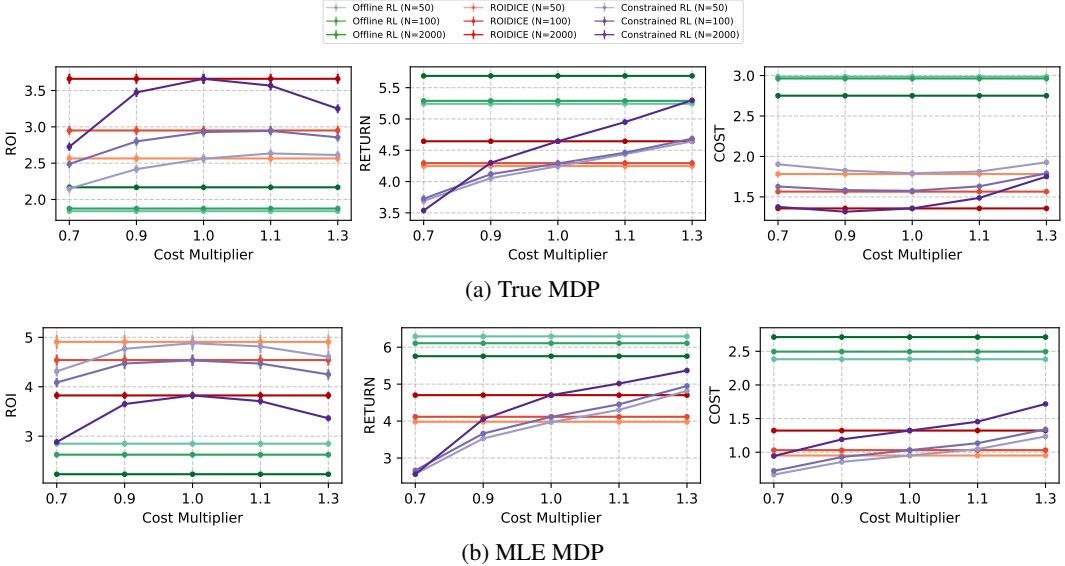

(a) True MDP

(b) MLE MDP

Figure 4: ROIDICE vs. offline constrained RL. We average the scores and obtain $\pm 2\times$ standard error using 1000 seeds. $N$ denotes the number of trajectories within the dataset.

## C  Details of Random MDP experiment

We evaluate ROIDICE on random MDP environments and compare with unconstrained and constrained RL algorithms, OptiDICE and COptiDICE. We generate tabular environments with 1000 different seeds. Each environments consists of 50 states and 4 actions. Their transition probability is randomly generated following $Dirichlet$ distribution. The sparse reward is given when an agent reaches on the goal state, and the cost function follows $Beta(0.2, 0.2)$ distribution. We add 1 to all states in order to satisfying the domain condition of linear-fractional programming.

$$r(s, a) = \begin{cases} \frac{10}{1-\gamma} & \text{if } s \text{ is the goal state} \\ 0 & \text{otherwise} \end{cases}$$

$$c(s, a) = \begin{cases} 0 & \text{if } s \text{ is the goal state} \\ 10 \cdot \beta + 1 & \text{otherwise} \end{cases}$$

where the discount factor $\gamma = 0.95$ and $\beta \sim Beta(0.2, 0.2)$.

After the MDP wih a cost function is generated, we obtain behavior policy $\pi_D(a|s)$ that achieves 90% of the optimal return to gather fixed datasets with $\{10, 20, 50, 100, 1000, 2000\}$ trajectories. The dataset is used to estimate the empirical transition probability $\widehat{T}(s'|s, a)$ with the given samples.

With the estimated transition probability $\widehat{T}(s'|s, a)$ and behavior policy $\pi_D(a|s)$, we apply the estimated MDP to a tabular version of ROIDICE. For the tabular version of ROIDICE, we apply the primal problem of the Regularized ROI maximization framework to a convex optimization solver software. For the convex regularization $f(x, t)$, we use $f_1(x, t) = \frac{1}{2}(x - t)^2$ and $\alpha$ is set to be the inverse of the number of trajectories.

$$\max_{d' \geq 0, t \geq 0} \sum_{s,a} d'(s, a) r(s, a) - \alpha \sum_{s,a} d_D(s, a) f\left(\frac{d'(s, a)}{d_D(s, a)}, t\right)$$

$$\text{s.t. } (\mathcal{B}_* d')(s) = t(1 - \gamma) p_0(s) + \gamma(\widehat{\mathcal{T}}_* d')(s) \ \forall s \tag{22}$$

$$\sum_{s,a} d'(s, a) c(s, a) = 1 \tag{23}$$

where $(\widehat{\mathcal{T}}_* d)(s) = \sum_{\bar{s}, \bar{a}} \widehat{T}(s|\bar{s}, \bar{a}) d(\bar{s}, \bar{a})$.

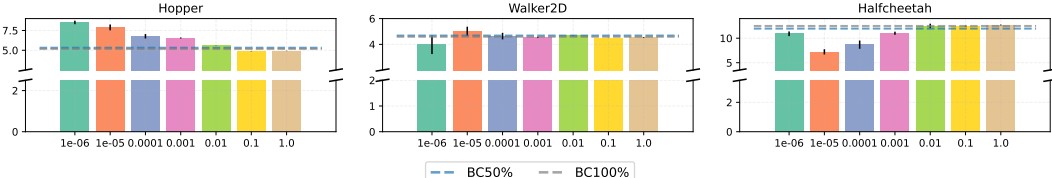

Figure 5: Comparison of different levels of the hyperparameter ($\alpha$) of ROIDICE in locomotion environments using expert data quality. We average the scores and obtain $\pm 2\times$ standard error using 5 seeds across 10 episodes.

## D  Impact of Various $\alpha$

We investigate how varying $\alpha$, which controls the strength of the regularization, affects the performance of ROIDICE by sweeping its values. Figure 5 shows the ROI performance for $\alpha$ ranging from $1 \times 10^{-6}$ to $1.0$ using expert quality dataset. Recall that the Hopper task datasets generally contain trajectories with low ROI, while most trajectories of Walker2d and Halfcheetah achieve relatively high ROI (see Table 2). In the Hopper environment, ROI performance decreases as $\alpha$ increases, indicating an opportunity to increase ROI when the model is less regularized, as the behavior policy collecting the offline dataset mostly does not account for the cost. In the Halfcheetah environment, ROIDICE achieves a high ROI when following the offline dataset due to the dataset's properties, but shows largely varying performance with small $\alpha$ due to its lack of coverage outside the dataset. In the Walker2D environment, ROIDICE shows robustness across various levels of $\alpha$.

## E  Data and Environment Details

We assess ROIDICE across four environments: three locomotion tasks (Hopper, Halfcheetah, Walker2D) and one financial trading task (FinRL). We utilize environment configurations and datasets provided by [6] (CC BY 4.0) and [18] (CC BY 4.0), with modifications to reward and cost functions. Additionally, we incorporate an absorbing state and an absorbing action to precisely calculate the sum of discounted rewards and costs, including an additional dimension to indicate whether the current state and action are absorbed.

**Locomotion Task**  In the Hopper environment, the reward is multifaceted, consisting of two primary components: the forward reward and the control cost. The forward reward incentivizes the agent to make progress towards its goal, while the control cost penalizes actions that are overly aggressive or energetically inefficient. Halfcheetah and Walker2D environments introduce an additional aspect to the reward: a healthy reward. This rewards the agent for maintaining stability. The control cost serves as a mechanism to penalize excessive actions, reflecting the idea that in real-world scenarios, significant force often comes with considerable energy expenditure. This cost can be seen as proportional to the energy consumption in physical robots, motivating agents to find smoother and more energy-efficient strategies for accomplishing their tasks. We utilize the forward reward as a reward and the control cost as a cost, normalizing the cost for stable learning. The reward and cost functions are as follows:

$$r(s,a) = forward\_reward\_weight * \frac{x_t - x_{t+1}}{dt}$$
$$c(s,a) = \frac{w_c^m}{|A|} \cdot \|a\|_2^2 + b_c^m$$

where $x_t$ represents the x-coordinate before the action $a_t$, and $x_{t+1}$ represents the x-coordinate after the action. The parameter $forward\_reward\_weight$ corresponds to the forward reward and is specific to each environment. The two hyperparameters, $w_c > 0$ and $b_c > 0$, are weights that control the strength of the cost, ensuring compliance with the linear-fractional program condition. Additionally, $|A|$ denotes the size of the action space dimension.

**Financial Task**  FinRL offers a stock trading simulation environment where agents trade stocks to earn compensation, with 30 tradable stocks available ($N = 30$). The reward is calculated by

summing the current balance with the product of the number of shares owned and the closing price of each share. Additionally, a cost function considers trading volume and stock price, imposing a fee on each transaction. The reward and cost functions are as follows:

$$r(s, a, s') = w_r^f \cdot (v(s') - v(s))$$

$$v(s) = s[0] + \sum s[1:31] * s[31:61]$$

$$c_i(s, a) = b_c^f + w_c^f \begin{cases} s[i+1] \cdot \min(a_i, s[i+N+1] & \text{if } a_i < 0 \\ s[i+1] \cdot \min(\frac{s[0]}{s[i+1]}, a_i) * pct & \text{if } a_i > 0 \end{cases}$$

where $pct = 0.001$ denotes a per-share percentage, which is the most commonly used transaction cost rate for each trade, $i = 0, 1, \ldots, N$ represents the index of tradable stocks, and $s[j], j \in \{0, \ldots, 181\}$, represents $j$-th dimension of state $s$. Each state dimension contains different information: $s[0]$ represents the balance, $s[1:31]$ denotes today's closing value for each of the 30 stocks, and $s[31:61]$ indicates today's trading volume for each of the 30 stocks.

Table 2: Reward and cost function hyperparameters

| Hyperparameter | Value |
|----------------|-------|
| $w_c^m$ | 1.0 |
| $b_c^m$ | 0.1 |
| $w_r^f$ | 0.001 |
| $w_c^f$ | 0.0001 |
| $b_c^f$ | 0.1 |

## F   Implementation Details

We implement ROIDICE algorithm based on [13] (CC BY 4.0) for the tabular experiment, and [20] (MIT License) for the continuous domain experiment. Our code is available at: `https://github.com/ku-dmlab/ROIDICE`. The implementation and configuration of OptiDICE and COptiDICE are adopted from the framework presented in [13], while CDT is utilized within the OSRL framework as described in [14].

The task and dataset name is succinctly stated: Hopper (H), Walker2D (W), Halfcheetah (HC), and Finance (F); for locomotion tasks, medium (m), medium-expert (m-e), and expert (e); for the financial task, medium (M), and high (H). We set the hyperparameter $\alpha$ for OptiDICE and COptiDICE, which controls the strength of the f-divergence regularizer. OptiDICE and COptiDICE have similar regularizer strengths, so we set the same $\alpha$ for both of them. For Hopper, we set $\alpha = 0.01$, for Walker2D (m) $\alpha = 0.1$, for Walker2D (m-e and e) $\alpha = 1.0$, and for Halfcheetah (respectively, m, m-e, and e) we set $\alpha = 0.01$, $\alpha = 0.1$, and $\alpha = 1.0$; for financial task we set $\alpha = 1.0$. For hyperparameter settings and model parameter configuration of ROIDICE, see Table 3

Table 3: Hyperparameters and configuration of ROIDICE.

| Task | | Value |
|---|---|---|
| Locomotion | $\alpha$ | 1e-06 (H-m, H-e), 1e-05 (W-e, HC-m), 1e-04 (H-m-e, W-m), 1e-03 (W-m, HC-m-e), 1.0 (HC-e) |
| | hidden dim of $\nu$ | MLP (256, 256) |
| | hidden dim of $\pi$ | MLP (256, 256) |
| | learning rate | 3e-04 |
| | $\gamma$ | 0.99 |
| | optimizer | Adam |
| Finance | $\alpha$ | 1.0 (F-M), 0.001 (F-H) |
| | hidden dim of $\nu$ | MLP (256, 256, 256) |
| | hidden dim of $\pi$ | MLP (256, 256, 256) |
| | learning rate | 3e-04 |
| | $\gamma$ | 0.99 |
| | optimizer | Adam |

## G Experiments Compute Resources

ROIDICE was conducted on a single CPU (AMD Ryzen Threadripper PRO 5995WX) with 256GB of RAM and a single GPU (NVIDIA RTX 4090). Each training session took about 10 to 20 minutes, utilizing approximately 20% of the RAM and from 40% to 60% of the GPU memory.

Table 4: Comparison of the runtime and number of parameters between algorithms. All algorithms, including baseline methods, were trained for 100K iterations on a single NVIDIA RTX 4090 GPU.

| | ROIDICE (Ours) | OptiDICE | COptiDICE | CDT |
|---|---|---|---|---|
| Run time (wall-clock) locomotion | 10 min. | 8 min. | 35 min. | 150 min. |
| Run time (wall-clock) finance | 20 min. | 16 min. | 120 min. | 250 min. |
| The number of parameters | 140K | 140K | 357K | 730K |

## H Experiments Results

We average the scores and obtain $\pm 2\times$ standard error using 5 seeds across 10 episodes. The task names are concisely abbreviated as follows: Hopper (H), Walker2D (W), HalfCheetah (HC), and Finance (F). Figure 6 illustrates the learning curves of ROIDICE and other offline unconstrained and constrained RL algorithms. In the Hopper environment, the performance of ROIDICE is significantly better. OptiDICE attains excessive returns but relatively lower ROI than ROIDICE due to its disregard for costs in the environment (see Table 5). ROIDICE achieves greater returns than OptiDICE when dealing with datasets that contain a mix of expert-level and lower-level data. Our analysis shows that ROIDICE exhibits qualitative behavior by considering both return and accumulated cost. Offline constrained RL algorithms, COptiDICE and CDT, show high ROI with datasets that include high ROI trajectories. In the HalfCheetah and Finance environments with high-quality datasets, CDT performs well but not as well as ROIDICE (see Table 6).

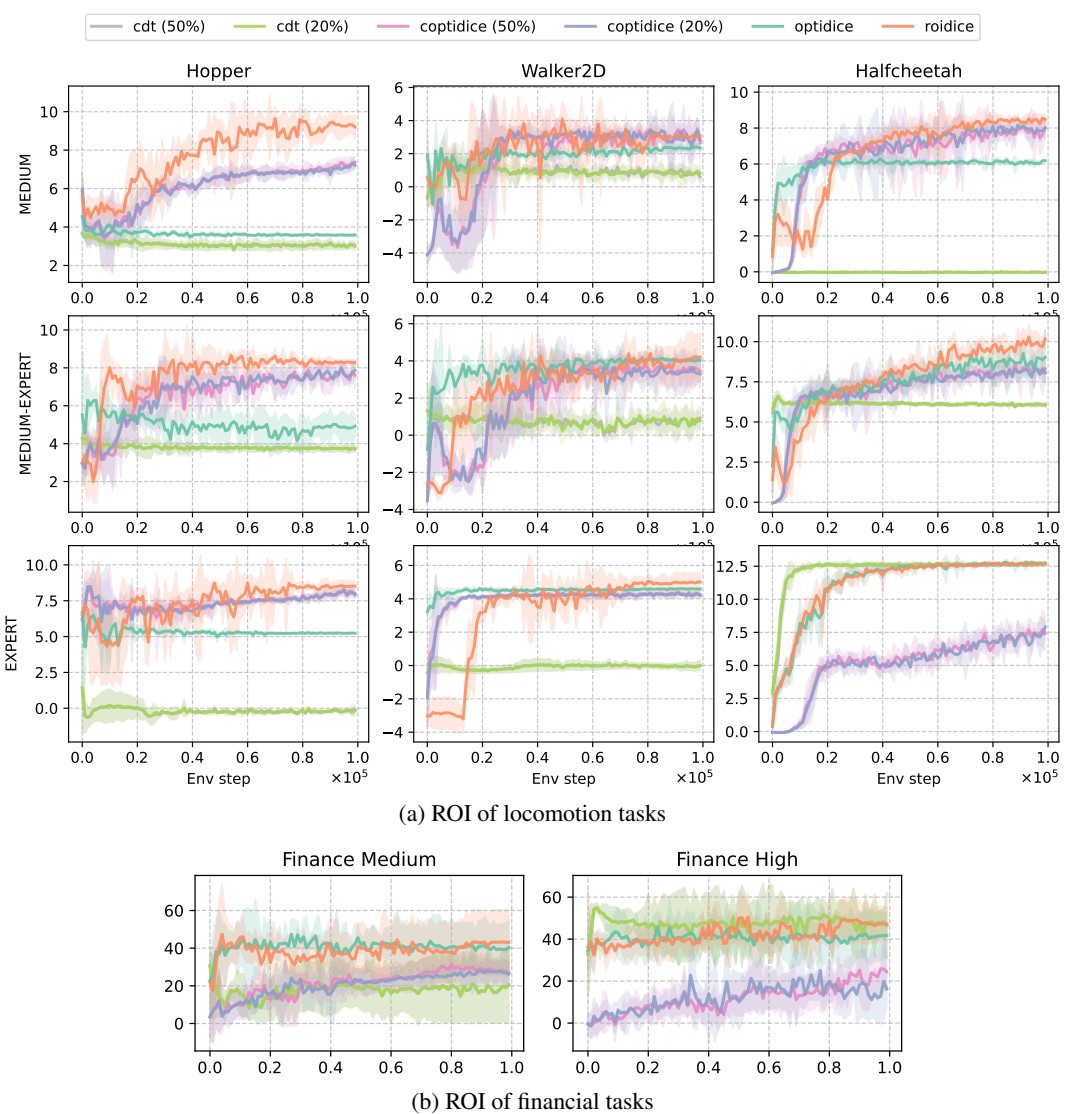

(a) ROI of locomotion tasks

(b) ROI of financial tasks

Figure 6: For each tasks, we report average ROI, return, and cost return with $\pm 2\times$ standard error with 5 seeds across 10 episodes.

Table 5: Results of ROIDICE compared with offline RL algorithms.

| Task | | ROIDICE | OptiDICE |
|------|------|---------|----------|
| H-m | $R_\pi$ | 102.414±10.19 | 134.61±0.64 |
| | $C_\pi$ | 11.124±0.98 | 37.61±0.24 |
| H-m-e | $R_\pi$ | 151.36±1.96 | 149.84±10.33 |
| | $C_\pi$ | 18.26±0.27 | 30.43±0.26 |
| H-e | $R_\pi$ | 115.11±12.24 | 171.95±1.19 |
| | $C_\pi$ | 13.52±1.28 | 37.61±0.24 |
| W-m | $R_\pi$ | 69.94±0.63 | 111.88±6.33 |
| | $C_\pi$ | 22.92±1.07 | 47.87±0.62 |
| W-m-e | $R_\pi$ | 148.71±27.95 | 192.13±2.46 |
| | $C_\pi$ | 35.35±1.12 | 47.68±0.7 |
| W-e | $R_\pi$ | 123.13±14.55 | 211.95±0.51 |
| | $C_\pi$ | 24.49±0.95 | 46.2±0.18 |
| HC-m | $R_\pi$ | 361.39±14.67 | 435.59±3.52 |
| | $C_\pi$ | 42.57±1.05 | 70.4±0.13 |
| HC-m-e | $R_\pi$ | 577.59±23.47 | 574.9±54.68 |
| | $C_\pi$ | 56.78±0.27 | 64.39±1.3 |
| HC-e | $R_\pi$ | 762.98±5.91 | 759.75±10.55 |
| | $C_\pi$ | 60.11±0.17 | 59.61±0.25 |
| F-M | $R_\pi$ | 1032.66±365.62 | 981.2±312.06 |
| | $C_\pi$ | 23.56±2.02 | 23.91±2.16 |
| F-H | $R_\pi$ | 1093.86±288.69 | 1076.16±157.41 |
| | $C_\pi$ | 23.05±2.64 | 25.03±2.3 |

Table 6: Results of offline constrained RL algorithms.

| Task | | COptiDICE | | CDT | |
| --- | --- | --- | --- | --- | --- |
| | | 50th | 80th | 50th | 20th |
| H-m | $R_\pi$ | 130.02±4.0 | 131.45±2.35 | 81.7±8.09 | 84.03±11.63 |
| | $C_\pi$ | 17.64±0.65 | 18.25±0.65 | 27.08±1.33 | 27.43±2.02 |
| H-m-e | $R_\pi$ | 93.2±22.9 | 103.23±14.57 | 126.3±7.62 | 127.86±5.54 |
| | $C_\pi$ | 12.14±2.42 | 13.1±1.72 | 34.23±1.69 | 33.99±2.06 |
| H-e | $R_\pi$ | 120.43±10.09 | 114.05±5.54 | -0.6±1.44 | -1.16±1.84 |
| | $C_\pi$ | 15.24±0.97 | 14.28±0.71 | 7.93±0.36 | 7.79±0.41 |
| W-m | $R_\pi$ | 75.38±19.38 | 79.6±13.44 | 30.0±20.79 | 40.04±17.1 |
| | $C_\pi$ | 29.12±0.95 | 29.08±0.56 | 48.87±2.58 | 49.74±3.14 |
| W-m-e | $R_\pi$ | 100.7±16.86 | 94.33±14.25 | 47.52±25.17 | 41.76±18.24 |
| | $C_\pi$ | 29.18±0.89 | 29.6±0.45 | 47.91±3.24 | 48.67±4.43 |
| W-e | $R_\pi$ | 120.7±9.7 | 117.91±9.45 | -0.03±5.07 | 0.29±5.45 |
| | $C_\pi$ | 28.38±1.64 | 28.03±1.67 | 24.96±0.39 | 24.98±0.37 |
| HC-m | $R_\pi$ | 229.85±15.37 | 297.36±14.02 | -0.26±0.12 | -0.11±0.13 |
| | $C_\pi$ | 37.07±0.48 | 37.21±0.58 | 10.0±0.0 | 10.0±0.0 |
| HC-m-e | $R_\pi$ | 323.48±18.3 | 320.2±16.21 | 432.69±8.78 | 423.53±4.56 |
| | $C_\pi$ | 38.3±1.32 | 38.76±0.94 | 70.37±0.35 | 70.71±0.63 |
| HC-e | $R_\pi$ | 235.13±28.85 | 255.62±16.18 | 768.67±4.38 | 768.64±4.41 |
| | $C_\pi$ | 31.04±0.78 | 31.67±0.84 | 60.66±0.65 | 60.71±0.58 |
| F-M | $R_\pi$ | 570.02±246.88 | 351.89±114.96 | 528.08±321.16 | 528.54±321.15 |
| | $C_\pi$ | 22.75±2.03 | 22.13±1.94 | 24.84±1.71 | 24.84±1.72 |
| F-H | $R_\pi$ | 665.92±146.82 | 639.37±107.21 | 1012.48±212.15 | 1012.51±212.17 |
| | $C_\pi$ | 24.5±0.95 | 24.21±0.45 | 21.68±0.91 | 21.68±0.91 |

# I Comparison of Offline Constrained RL Algorithms

Table 7: ROI of ROIDICE compared with offline constrained RL algorithms. We average each score and get $\pm 2\times$ standard error with 5 seeds across 10 episodes.

| Data quality | medium | medium-expert | expert |
|---|---|---|---|
| ROIDICE | 9.21±0.49 | 8.29±0.18 | 8.51±0.25 |
| VOCE 50th | 1.807±0.63 | 1.33±1.28 | 1.34±0.67 |
| VOCE 80th | 1.799±0.62 | 1.34±1.3 | 1.47±0.76 |
| CPQ 50th | 1.13±0.58 | -0.06±0.41 | -0.11±0.71 |
| CPQ 80th | -0.10±0.64 | -0.15±0.48 | -0.70±0.19 |

Our experiment in continuous domains from Section 5.2 includes non-DICE offline constrained algorithm, CDT. We additionally provide experiments on other offline constrained RL frameworks, VOCE [8] and CPQ [21]. Table 7 presents the ROI performance of these constrained offline RL algorithms in the Hopper environment [6]. These algorithms use two cost budgets, corresponding to the 50th and 80th percentiles of the accumulated costs from the offline dataset.

Over various experiments, constrained offline RL algorithms commonly have shown low ROIs. We conjecture that these results are mainly due to the overestimation of costs, making constrained offline RL algorithms to be overly conservative on choosing costly state-actions. On the other hand, ROIDICE seems to be less affected by such overestimation, as overestimated costs are partially mitigated with overestimated rewards when estimating ROIs.

# J Experiment Results in Safety RL Environments

Table 8: ROI of ROIDICE compared with offline constrained RL algorithms. We average each score and get $\pm 2\times$ standard error with 5 seeds across 10 episodes.

| Task | CarGoal | PointPush |
|---|---|---|
| ROIDICE | 0.160±0.02 | 0.033±0.01 |
| COptiDICE 50th | 0.119±0.01 | 0.026±0.01 |
| COptiDICE 80th | 0.115±0.01 | 0.022±0.01 |

We evaluate our approach on OpenAI SafetyGym [19] tasks including CarGoal and PointPush. Table 8 provides results averaged over 5 seeds across 10 episodes. We observed that ROIDICE outperforms COptiDICE in terms of ROI. We use the rewards and costs from the environment, adding a constant value of $\epsilon = 0.1$ to each cost to maintain our assumption that $c(s, a) > 0 \forall s, a$. The offline dataset is collected by PPO Lagrangian. ROIDICE uses $\alpha = 0.001$ for CarGoal and $\alpha = 0.01$ for PointPush. COptiDICE uses $\alpha = 0.01$ for CarGoal and $\alpha = 1.0$ for PointPush.

