# OpenReview forum: "ROIDICE: Offline Return on Investment Maximization for Efficient Decision Making"
_NeurIPS.cc/2024/Conference — NeurIPS 2024 poster_

### Official Review · Reviewer_Yyxa · 2024-07-02

**Soundness:** 3
**Presentation:** 3
**Contribution:** 2
**Rating:** 5
**Confidence:** 3

**Summary:**

The article proposes a novel offline algorithm for learning policies optimising Return on Investment (ROI): the ratio between the (discounted) cumulated rewards obtained by a policy, and the (discounted) cumulated costs of the actions taken. The paper focuses specifically on offline RL, and in particular builds on formalising the RL problem in a Linear Programming form.
The proposed approach builds on the recent literature, particularly DICE-RL, but derives a new regularisation term for ROI optimisation.

The experimental evaluation demonstrates that the proposed approach successfully learns policies with high ROI on various offline RL datasets.

**Strengths:**

- The idea of optimising ROI is, to my knowledge, novel, and interesting as it proposes a new way to balance the trade-off between action costs and policy rewards, which is relevant in a wide range of applications of RL.
- Although the RL literature has proposed ways to handle this trade-offs since its early days (for example, by modelling costs as negative rewards), there is an argument that an ROI objective could have some advantages.
- The derivation of the algorithm is well described in the article and is a substantive contribution with respect to the state-of-the-art

**Weaknesses:**

- Although optimising ROI could indeed provide a better framework for cost/returns tradeoffs, the article does not provide a substantive argument of why this would be the case. This somewhat reduces the expected impact and relevance of the contribution.
- Similarly, the experimental evaluation only compares the proposed approach to RL trained to optimise the return solely, with no constraint for cost. In this setting, it seems self-evident that the proposed method would achieve better ROI and lower costs than a policy trained ignoring cost altogether. It is fairly common in RL to approach this tradeoff by integrating the action costs as negative rewards, and it would have been fairer to compare with such an approach as an additional baseline.
- One reason for the broad use of ROI in business, is that the quantities of "return" and "investment" are usually unambiguously in the same quantity making the ratio calculation natural (although different costs can affect the ratio). This is less straightforward in many RL problems where the return and possible costs may be in very different dimensions and distributions. It is unclear from the paper how stable are the policies optimised depending on reward/cost scaling.
- There is an issue with a number of references in the bibliography that appear to be missing some fields (eg, refs 6,7,9,10,11)Could you provide a better rationale on why ROI would be a good objective for RL in general, beyond its use in the business domain?
- The difference in return between standard RL and the ROI version is quite large in the experiments. How significant is the decrease in policy performance compared to cost-free policies in practice for
- Does the proposed approach provide some guarantees on cost, compared to, eg, CoptiDICE?

**Questions:**

Could you provide a better rationale on why ROI would be a good objective for RL in general, beyond its use in the business domain?
- The difference in return between standard RL and the ROI version is quite large in the experiments. How significant is the decrease in policy performance compared to cost-free policies in practice for
- Does the proposed approach provide some guarantees on cost, compared to, eg, CoptiDICE?

**Limitations:**

- No significant ethical aspects to this work.

---

> ### Author Rebuttal · Authors · 2024-08-07
>
> Dear Reviewer Yyxa,
>
> We appreciate your feedback and have provided responses to your questions below.
>
> **1. Could you provide a substantive argument of why optimizing ROI could offer a better framework for cost/returns trade-offs?**
>
> Constrained reinforcement learning is a framework that aims to maximize the return within a given cost budget, rather than optimizing the trade-off between returns and costs. In other words, constrained RL agent does not try to minimize the accumulated cost if it is below the budget. However, ROI maximization focuses on maximizing the return/cost ratio, meaning the accumulated cost can be minimized further to achieve a higher ROI.
>
> If our main goal is to maximize ROI, ROIDICE is much more efficient compared to utilizing constrained RL algorithms. As described in Section 5.1, for ROI maximization, constrained RL requires multiple runs with different levels of cost budgets, and the policy with the best ROI performance can be selected afterward. However, ROIDICE requires only a single run to obtain the policy with the best ROI performance, eliminating the need to search among cost budget candidates.
>
> **2. It is fairly common in RL to approach cost/return trade-off by integrating the action costs as negative rewards.**
>
> In our paper, ROIDICE has been compared with offline RL and offline constrained RL. As the reviewer has pointed out, the return/cost trade-off can be optimized with a regularized RL framework with its reward $r(s,a)-\lambda c(s,a)$. However, we believe that constrained RL is a better baseline for ROIDICE when considering practical usages.
>
> Theoretically, there always exists a regularized RL with a certain $\lambda$ that provides the same solution as constrained RL for a given cost budget. In some constrained RL algorithms using Lagrangian, $\lambda$ is implicitly optimized to satisfy the cost constraint[1].
> However, finding an appropriate $\lambda$ to compare regularized RL methods with ROIDICE can be difficult, particularly when the scale of reward and cost varies significantly. In contrast, constrained RL is more practical, as a suitable cost budget can be inferred from the dataset, alleviating the need for an explicit search for $\lambda$.
>
> **3. How stable are the policies optimized in problems where the returns and possible costs may have very different dimensions and distributions?**
>
> As mentioned in the Limitations section, ROIDICE can be influenced by the design of reward and cost functions. The variation of reward and cost across state-action pairs significantly impacts ROIDICE as it maximizes the ratio of return to accumulated cost. For instance, as noted in Section 3, when the cost is constant across all states and actions, the ROIDICE policy becomes equivalent to the standard offline RL policy. Conversely, as the distributions of cost become more diverse, the ROIDICE policy deviates from the offline RL policy.
>
> On the other hand, analogous to typical RL algorithms not being affected by the scales of reward function, ROIDICE is basically invariant to different scales of rewards and costs apart from the choice of appropriate $\alpha$. We also empirically observed that ROIDICE resulted in a same level of performance when $\alpha$ is appropriately scaled to align with the cost scales.
>
> **4. Could you provide a better rationale on why ROI would be a good objective for RL in general, beyond its use in the business domain?**
>
> We can come up with a number of practical scenarios where ROI would be a good objective, such as optimizing the mileage (km/L) of an autonomous vehicle, the (decision optimality/planning time) trade-off when adopting a meta controller for real-time decision making, or the speed-accuracy tradeoff (generation quality/inference time) of a large language model (LLM). We illustrate a comparison between two approaches for optimizing the mileage problem.
>
> - Constrained RL: An agent trained with constrained RL uses a limited budget to maximize the reward. Setting an appropriate budget that is feasible for the problem may require domain knowledge. Therefore, to compare the ROI performance of its policy, it is necessary to conduct multiple runs of constrained RL with different levels of constraints (e.g., fuel consumption $\leq$ [5, 10, 15, 20]). The policy with the best mileage is then selected from these multiple runs of constrained RL.
> - ROI maximization: A single run of ROIDICE is sufficient to find a policy that maximizes the vehicle's mileage.
>
> Ideally, the mileage (ROI) performance of the ROIDICE policy would be equal to or greater than that of the constrained RL policy. Equality occurs when the upper bound of the constraint matches the fuel consumption of the ROIDICE policy.
>
> **5. How significant is the decrease in policy performance compared to cost-free policies in practice for?**
>
> The remark in Section 3 notes that the optimal policies of standard RL and ROI maximization are equivalent when the cost is constant across all states and actions. This gap increases when the cost varies among states and actions or when there is an action with high cost that leads to high return. The difference between standard RL and ROIDICE can be managed by adjusting the cost distribution.
>
> **6. Does the proposed approach provide some guarantees on cost?**
>
> Current form of ROIDICE does not guarantee the upper bound on cost. However, ROIDICE with the cost guarantee can be easily derived by adding the cost constraint $\sum_{s,a}d(s,a)c(s,a)\leq C_{\text{threshold}}$ to the LP formulation of our algorithm, ROI-LP.
>
>
> Thanks for pointing out the issue with references. We will fix this in the next revision.
>
> [1] J. Lee, et al. COptiDICE: Offline constrained reinforcement learning via stationary distribution correction estimation. ICLR, 2022.

---

> > ### Comment · Reviewer_Yyxa · 2024-08-12
> > **Good rebuttal**
> >
> > Thank you for the detailed and thoughtful response to my comments. After reading the response (and other reviewers' comments), I would like to raise my rating to 6.

---

> > > ### Author Response · Authors · 2024-08-13
> > >
> > > Dear Reviewer Yyxa,
> > >
> > > We're pleased to know that your concerns have been resolved, and we truly appreciate your positive feedback!
> > >
> > > However, it seems the system is still displaying the initial review score. Could you kindly verify if your updated score has been correctly applied?
> > >
> > > Thank you so much.

---

> ### Comment · Area_Chair_dKSQ · 2024-08-11
> **Rebuttal discussion**
>
> Dear Reviewer Yyxa,
>
> Could you please take a look at the rebuttal and share your thoughts on the authors' reply? The author has provided a detailed response to the initial comments - does it address your major concerns?
>
> Thank you very much for your time and effort!
>
> Best,
>
> Your AC

---

### Official Review · Reviewer_T1wG · 2024-07-11

**Soundness:** 3
**Presentation:** 3
**Contribution:** 3
**Rating:** 6
**Confidence:** 4

**Summary:**

This paper introduces a novel approach for solving constrained offline reinforcement learning problems. The authors apply the Charnes-Cooper transformation to convert the linear-fractional programming into an equivalent linear programming problem, and draw inspiration from the DICE framework to maximize ROI under offline setting. Experimental results across finite and continuous domains demonstrate that the proposed ROIDICE achieves a better trade-off between return and accumulated cost.

**Strengths:**

* The paper's approach is mathematically sound, and the motivation is well elucidated.
* The experimental results conducted in various environments demonstrate its superiority over the baseline.

**Weaknesses:**

* Important baselines are missing: apart from DICE-based methods, comparisons should include other offline constrained RL methods.
* How is the proposed method implemented in continuous domains? The description lacks clarity; for instance, providing pseudocode would be helpful.
* In the FinRL task, it's intriguing why the unconstrained OptiDICE method performed well while other constrained baselines showed poor performance.

**Questions:**

- It is suggested to add the discussion of related works and compare them in experiments, such as [1,2].
- Can you explain why using constraint methods in the FinRL environment actually yields poor results?
- Can you provide experimental results on widely recognized safe RL environments such as SafetyGymnasium[3,4] and BulletSafetyGym[5]?
  [1] Xu H, Zhan X, Zhu X. Constraints penalized q-learning for safe offline reinforcement learning[C]//Proceedings of the AAAI Conference on Artificial Intelligence. 2022, 36(8): 8753-8760.
  [2] Guan J, Chen G, Ji J, et al. VOCE: Variational optimization with conservative estimation for offline safe reinforcement learning[J]. Advances in Neural Information Processing Systems, 2024, 36.
  [3] Ray A, Achiam J, Amodei D. Benchmarking safe exploration in deep reinforcement learning[J]. arXiv preprint arXiv:1910.01708, 2019, 7(1): 2.
  [4] Ji J, Zhou J, Zhang B, et al. Omnisafe: An infrastructure for accelerating safe reinforcement learning research[J]. arXiv preprint arXiv:2305.09304, 2023.
  [5] Gronauer S. Bullet-safety-gym: A framework for constrained reinforcement learning[J]. 2022.

**Limitations:**

As mentioned in the paper, the ultimate performance of ROIDICE is contingent upon the dataset and exhibits sensitivity, potentially resulting in behaviors such as minimizing costs excessively.

---

> ### Author Rebuttal · Authors · 2024-08-07
>
> Dear Reviewer T1wG,
>
> We thank the reviewer for the detailed feedback. We address your remarks below.
>
> **1. Important baselines are missing: apart from DICE-based methods, comparisons should include other offline constrained RL methods.**
>
> Our experiment in continuous domains from Section 5.2 includes non-DICE offline constrained algorithm, CDT[1]. We additionally provide experiments on other offline constrained RL frameworks, VOCE[2] and CPQ[3]. Table 1 in the rebuttal supplementary presents the ROI performance of these constrained offline RL algorithms in the Hopper environment[4]. These algorithms use two cost budgets, corresponding to the 50th and 80th percentiles of the accumulated costs from the offline dataset.
>
> Over various experiments, constrained offline RL algorithms commonly have shown low ROIs. We conjecture that these results are mainly due to the overestimation of costs, making constrained offline RL algorithms to be overly conservative on choosing costly state-actions. On the other hand, ROIDICE seems to be less affected by such overestimation, as overestimated costs are partially mitigated with overestimated rewards when estimating ROIs.
>
> **2. How is the proposed method implemented in continuous domains?**
>
> We provide the pseudocode of ROIDICE in the rebuttal supplementary. Lagrangian multipliers are parameterized and updated to optimize ROIDICE objectives presented in our paper. Optimal policy $\pi^*_\theta$ is extracted from the optimal ratio $w^{*}_{\nu,\mu,t}$ obtained from the Lagrangian multipliers.
>
> **3. In the FinRL task, why the unconstrained RL performed better than constrained RL and what makes the constrained RL methods perform poorly?**
>
> As reported in the appendix, in FinRL task, constrained RL algorithms are trying to achieve lower cost compared to unconstrained RL algorithms by sacrificing large amount of its return. While both constrained and unconstrained RL agents are resulting in a policy with smaller cost compared to the cost budget, we assume that constrained RL algorithms are largely overestimating the cost, resulting in a overly conservative policies.
>
> **4. Can you provide experimental results on widely recognized safe RL environments?**
>
> We evaluate our approach on OpenAI SafetyGym[5] tasks including CarGoal and PointPush. Table 2 in the rebuttal supplementary materials provides results averaged over 5 seeds across 10 episodes. Similar to the reported results in the paper, we observed that ROIDICE outperforms COptiDICE in terms of ROI. We use the rewards and costs from the environment, adding a constant value of $\epsilon=0.1$ to each cost to maintain our assumption that $c(s,a) > 0 \ \forall s,a$. The offline dataset is collected by PPO Lagrangian. ROIDICE uses $\alpha=0.001$ for CarGoal and $\alpha=0.01$ for PointPush. COptiDICE[6] uses $\alpha=0.01$ for CarGoal and $\alpha=1.0$ for PointPush. We will include the results in the final version of the paper if accepted.
>
> [1] Z. Liu, et al. Constrained decision transformer for offline safe reinforcement learning. ICML, 2023.
>
> [2] J. Guan et al. VOCE: Variational optimization with conservative estimation for offline safe reinforcement learning. NeurIPS, 2024.
>
> [3] H. Xu, et al. Constraints penalized q-learning for safe offline reinforcement learning. AAAI, 2022.
>
> [4] J. Fu, et al. D4rl: Datasets for deep data-driven reinforcement learning, 2020.
>
> [5] A. Ray, et al. Benchmarking safe exploration in deep reinforcement learning. arXiv preprint arXiv:1910.01708, 2019.
>
> [6] J. Lee, et al. COptiDICE: Offline constrained reinforcement learning via stationary distribution correction estimation. ICLR, 2022.

---

> > ### Comment · Area_Chair_dKSQ · 2024-08-11
> > **Rebuttal discussion**
> >
> > Dear Reviewer T1wG,
> >
> > Could you please take a look at the rebuttal and share your thoughts on the authors' reply? The author has provided additional comparisons to baselines, explanations, and more experimental results. Does the rebuttal address your initial concerns?
> >
> > Thank you very much for your time and effort!
> >
> > Best,
> >
> > Your AC

---

> > ### Comment · Reviewer_T1wG · 2024-08-12
> > **Good rebuttal that addressed most of my concerns**
> >
> > I thank the authors for providing detailed responses to my comments. Reading the rebuttal materials solved most of my concerns. I also read the comments from other reviewers and still rate this work positively.
> >
> > Good luck to the authors for the final decision of this work.
> >
> > Best wishes,

---

### Official Review · Reviewer_8zrU · 2024-07-12

**Soundness:** 3
**Presentation:** 3
**Contribution:** 2
**Rating:** 6
**Confidence:** 3

**Summary:**

This paper provides a fraction linear programming framework for solving offline RL problems for return on investment (ROI). The fraction linear programming can be trasformed to a linear programming and a convex regularizer is used to control the distribution mismatch. The authors provide adequate experimental results to demonstrate the advantages of the algorithm.

**Strengths:**

The framework is clean which uses linear programming to model the offline ROI maximization problems and the authors provide sufficient experimental results with good performance.

**Weaknesses:**

No sample complexity guarantee.  See questions for details.

**Questions:**

1. Can the authors provide theoretical analysis for the sample complexity like the paper  ``Ozdaglar et al Revisitting the linear programming framework for offline reinforcement learning'' which also uses LP framework to solve offline RL?
2. How can we implement this LP using function approximation when the state-action space is large?
3. Missing references: There are papers using LP frameworks to solve offline RL with sample complexity guarantee, such as the two paper mentioned before.

---

> ### Author Rebuttal · Authors · 2024-08-07
>
> Dear Reviewer 8zrU,
>
> We appreciate your comprehensive comments. Please find the response to your questions below.
>
> **1. Theoretical analysis for the sample complexity in ROI-LP.**
>
> Sample complexity analyses in the LP formulation of RL traditionally estimate a policy's return using a linear combination of its stationary distribution and reward. On the other hand, ROI is defined as a ratio between the linear combinations of the stationary distribution with reward and cost, making existing approaches not directly applicable.
>
> To illustrate this, we demonstrate that the method proposed in [1] cannot be directly applied to determine the sample complexity of the ROI-LP. Specifically, Lemma 3 in [1] cannot be straightforwardly derived from the ROI-LP. The lemma assumes an approximated stationary distribution ${\theta}\in \mathbb{R}^{|S||A|}$ and policy $\pi_{\theta}$ extracted from $\theta$. $d_{\pi_{\theta}}\in \mathbb{R}^{|S||A|}$ is the true stationary distribution generated by policy $\pi_{\theta}$. Lemma 3 provides an error bound that relates the Bellman flow constraint violation of $\theta$ and the absolute difference between $\sum_{s,a}\theta(s,a)r(s,a)$ and $\sum_{s,a}d_{\pi_\theta}(s,a)r(s,a)$.
>
>  **Lemma 3** For any $\theta\geq0$ and $r(s,a)\in[0,1]$, we have $|\sum_{s,a}r(s,a)(\theta(s,a)-d_{\pi_{\theta}}(s,a))|\leq{||M\theta -(1-\gamma)p_{0}||_{1} \over 1-\gamma}$,
>
> where $M=B_{\*}- \gamma T_{\*}$.
>
> We show that a straightforward extension of the lemma is not feasible in the ROI-LP context. The pair $(\tilde{\theta},\tilde{t})$ serves as an approximation of $(d^{'},t)$ in ROI-LP, and the policy $\pi_{\tilde{\theta}}$ is extracted from $\tilde{\theta}/\tilde{t}$. The actual stationary distribution produced by $\pi_{\tilde{\theta}}$ is $d_{\pi_{\tilde{\theta}}}^{'} / t_{\pi_{\tilde{\theta}}}$. We begin by assessing the extent to which $(\tilde{\theta},\tilde{t})$ violates the scaled Bellman flow constraint in ROI-LP.
>
> $\Vert M\tilde{\theta}-(1-\gamma)p_{0}\tilde{t}\Vert_{1} = \Vert M\tilde{\theta}-M\tilde{t}d_{\pi_{\tilde{\theta}}}^{'} / t_{\pi_{\tilde{\theta}}}\Vert_{1}\nonumber\geq(1-\gamma)\Vert B_{\*}\tilde{\theta}-B_{\*}d_{\pi_{\tilde{\theta}}}^{'} \tilde{t}/t_{\pi_{\tilde{\theta}}}\Vert_{1}$
>
> where a valid stationary distribution $d_{\pi_{\tilde{\theta}}}^{'} / t_{\pi_{\tilde{\theta}}}$ satisfies $(1-\gamma)p_0 = Md_{\pi_{\tilde{\theta}}}^{'} / t_{\pi_{\tilde{\theta}}}$.
>
> The ROI difference between $\sum_{s,a}\tilde{\theta}(s,a)r(s,a)$ and $\sum_{s,a}d_{\pi_{\tilde{\theta}}}^{'}(s,a)r(s,a)$ can be bounded by,
>
> $|\sum_{s,a}r(s,a)(\tilde{\theta}(s,a) - d_{\pi_{\tilde{\theta}}}^{'}(s,a))|\leq\sum_{s}|\sum_a \tilde{\theta}(s,a) - d_{\pi_{\tilde{\theta}}}^{'}(s,a)|=\Vert B_{\*}\tilde{\theta}-B_{\*}d_{\pi_{\tilde{\theta}}}^{'}\Vert_{1}$
>
> It can be noted that, due to the additional factor $\tilde{t} / t_{\pi_{\tilde{\theta}}}$ that cannot be easily bounded under finite samples, ROI-LP cannot use the same proof technique to [1] for sample complexity analysis.
>
> While the sample complexity analysis is indeed an important research topic, we believe that conducting a new type of sample complexity analysis appropriate for the Linear-Fractional Programming framework is beyond of the scope of this paper.
>
> [1] Ozdaglar, Asuman E., et al. "Revisiting the linear-programming framework for offline rl with general function approximation." International Conference on Machine Learning. PMLR, 2023.

---

> > ### Comment · Area_Chair_dKSQ · 2024-08-11
> > **Rebuttal discussion**
> >
> > Dear Reviewer 8zrU,
> >
> > Could you please take a look at the rebuttal and share your thoughts on the authors' reply? The author has provided additional theoretical analysis and clarification in response to Q1. Does the rebuttal sway your opinions?
> >
> > Thank you very much for your time and effort!
> >
> > Best,
> >
> > Your AC

---

> > ### Comment · Reviewer_8zrU · 2024-08-13
> >
> > I thank the authors for the response. I will increase the score to 6.  But I still think for offline RL papers, the sample complexity and function approximation setting to deal with large state action space are important.

---

### Official Review · Reviewer_Z94y · 2024-07-12

**Soundness:** 2
**Presentation:** 3
**Contribution:** 2
**Rating:** 5
**Confidence:** 2

**Summary:**

The paper addresses the problem of maximizing the Return on Investment (ROI) in the context of offline reinforcement learning (RL). The method introduced is ROIDICE, which stands for Return on Investment Decision-making in the Offline Setting. ROIDICE is an offline policy optimization algorithm designed to optimize the ROI of a policy using a fixed dataset of pre-collected experiences.
The method involves formulating the ROI maximization problem as a linear-fractional programming problem, which is then converted into an equivalent linear programming problem using the Charnes-Cooper transformation. This transformation allows the problem to be solved using standard linear programming techniques.
To address the distribution shift inherent in offline learning, ROIDICE incorporates a convex regularization that measures the amount of distribution shift while maintaining the convexity of the problem. This regularization is designed to penalize the trained policy's deviation from the behavior policy used to collect the offline dataset.
The experiments conducted in the paper demonstrate that ROIDICE outperforms other offline RL algorithms, including those that focus solely on maximizing the return of the policy. ROIDICE achieves a superior trade-off between return and accumulated cost, resulting in a more efficient policy.

**Strengths:**

1. The paper introduces a novel policy optimization framework that maximizes the Return on Investment (ROI) of a policy. This is a significant departure from traditional approaches that focus solely on maximizing return, without considering the cost associated with the actions.
2. The proposed method operates within an offline setting, where direct interaction with the environment is not possible. This is particularly useful in scenarios where online interaction is costly or risky.
3. The authors derive an offline algorithm, ROIDICE, which optimizes the ROI of a policy using a fixed dataset of pre-collected experiences.
4. The paper demonstrates that ROIDICE yields policies with better efficiency than policies from existing RL-based optimization methods. This is achieved by considering the trade-off between return and accumulated cost. The authors conduct extensive experiments across various domains, including locomotion tasks and a financial stock trading task. T

**Weaknesses:**

ROIDICE may require significant computational resources, especially when dealing with large-scale datasets and complex environments. Could you provide details on the computational resources and time consumption for different methods?
If the data distribution in the offline dataset significantly differs from that encountered in the online environment, the performance of ROIDICE may be affected. How can this issue be addressed?

**Questions:**

See Weakness

---

> ### Author Rebuttal · Authors · 2024-08-07
>
> Dear Reviewer Z94y,
>
> We thank the reviewer for the thorough and constructive comments. We hope we can address your concerns below.
>
> **1. Could you provide details on the computational resources and time consumption for different methods**
>
> We provide details on the resources and runtime to demonstrate that ROIDICE does not require a significant amount of resources for large-scale domains, such as locomotion and finance. All algorithms, including baseline methods, were trained for 100K iterations on a single NVIDIA RTX 4090 GPU.
>
> ||ROIDICE (Ours)|OptiDICE[1]|COptiDICE[2]|CDT[3]|
> |---|---|---|---|---|
> |Run time (wall-clock) locomotion|10 min.|8 min.|35 min.|150 min.|
> |Run time (wall_clock) finance|20 min.|16 min.|120 min.|250 min.|
> |# of parameters|140K|140K|357K|730K|
>
> **2. What if the data distribution in the offline dataset significantly differs from that encountered in the online environment?**
>
> In the context of offline reinforcement learning (RL), a major challenge arises from the degradation in performance due to increasing discrepancies between the offline data distribution and the online environment. To mitigate this issue, DICE-based RL frameworks—such as ROIDICE, OptiDICE, and COptiDICE—employ f-divergence to regulate the extent of conservativeness, which is modulated by the hyperparameter $\alpha$. As illustrated in Figure 5 of Appendix D, which corresponds to Figure 1 in the supplementary materials, selecting an appropriate $\alpha$ can identify an offline RL policy that is less affected by distributional shift.
>
> [1] J. Lee, et al. Optidice: Offline policy optimization via stationary distribution correction estimation. ICML, 2021.
>
> [2] J. Lee, et al. COptiDICE: Offline constrained reinforcement learning via stationary distribution correction estimation. ICLR, 2022.
>
> [3] Z. Liu, et al. Constrained decision transformer for offline safe reinforcement learning. ICML, 2023.

---

> > ### Comment · Area_Chair_dKSQ · 2024-08-11
> > **Rebuttal discussion**
> >
> > Dear Reviewer Z94y,
> >
> > Could you please take a look at the rebuttal and share your thoughts on the authors' reply? The author provided additional results on the computational cost and discussion on the online setting. Do they address your concerns?
> >
> > Thank you very much for your time and effort!
> >
> > Best,
> >
> > Your AC

---

### Author Rebuttal · Authors · 2024-08-07

We sincerely thank all the reviewers for their time and effort in providing constructive feedback and insightful reviews of our paper. We are grateful that the reviewers recognized our paper for presenting a novel offline policy optimization framework that effectively optimizes the trade-off between return and accumulated cost, is mathematically well-derived, and yields policies with better efficiency across various domains.

In response to the valuable feedback from the reviewers, we address each concern individually. Below, we summarize the major points of the feedback:
- How to address the discrepancy between the offline data distribution and the online environment.
- Whether a theoretical analysis of sample complexity is available for ROI-LP.
- Comparative analysis of ROI performance across policy optimization approaches
- Clarifying the advantages of ROIDICE for return/cost trade-off optimization
- Exploring potential applications of ROIDICE beyond business domains.

---

### Author Response · Authors · 2024-08-11
**Dear AC and PC,**

As we have now passed the midpoint of the author-reviewer discussion period, we would like to confirm whether our responses have effectively addressed the reviewers' concerns.

Could you kindly encourage the reviewers to begin their discussions? We would greatly appreciate your support in facilitating this process.

---

### Decision · Program_Chairs · 2024-09-25

**Decision:**

Accept (poster)

**Comment:**

The paper presents a constrained offline policy optimization algorithm that maximizes Return On Investment (ROI) and formulates it as a linear programming problem. The experiments on various offline RL tasks show that it achieves a better trade-off between return and accumulated cost.

The paper initially received borderline reviews. While the reviewers considered the work novel (Z94y, T1wG, Yyxa) and the proposed framework well-formulated (8zrU, T1wG, Yyxa), they also raised several concerns on the paper: 1) missing computational complexity analysis (Z94y) and theoretical analysis on sample complexity (8zrU) or cost guarantee (Yyxa); 2) lack of clarity on several technical issues, including coping with online setting (Z94y) and large state space (8zrU, T1wG), reward/cost scaling and advantage of the method (Yyxa) ; 3) missing important baselines and experiments (T1wG, Yyxa); 4) limited application domains (Yyxa); and 5) insufficient analysis on the results (T1wG, Yyxa).

The authors provided a thorough response, including additional results on computational complexity and experimental comparisons, detailed analysis of the theoretical properties and results, and clarifications on the technical issues and application domains. During the rebuttal discussion, most reviewers considered their main concerns well addressed and reached a consensus with positive ratings. After considering the paper, reviews, and rebuttal discussion, the AC does not find sufficient cause to overturn the reviewers' consensus and therefore recommends it for acceptance. The author should take into account the rebuttal and the reviewers' feedback, especially regarding the additional results and analysis, when preparing the final version.